# TGFβ reprograms TNF stimulation of macrophages towards a non-canonical pathway driving inflammatory osteoclastogenesis

Yuhan Xia[1,2,7,8], Kazuki Inoue[1,2,8], Yong Du [1], Stacey J. Baker[3], E. Premkumar Reddy[3], Matthew B. Greenblatt [4,5] & Baohong Zhao [1,2,6 ✉]

It is well-established that receptor activator of NF-κB ligand (RANKL) is the inducer of physiological osteoclast differentiation. However, the specific drivers and mechanisms driving inflammatory osteoclast differentiation under pathological conditions remain obscure. This is especially true given that inflammatory cytokines such as tumor necrosis factor (TNF) demonstrate little to no ability to directly drive osteoclast differentiation. Here, we found that transforming growth factor β (TGFβ) priming enables TNF to effectively induce osteoclastogenesis, independently of the canonical RANKL pathway. Lack of TGFβ signaling in macrophages suppresses inflammatory, but not basal, osteoclastogenesis and bone resorption in vivo. Mechanistically, TGFβ priming reprograms the macrophage response to TNF by remodeling chromatin accessibility and histone modifications, and enables TNF to induce a previously unrecognized non-canonical osteoclastogenic program, which includes suppression of the TNF-induced IRF1-IFNβ-IFN-stimulated-gene axis, IRF8 degradation and B-Myb induction. These mechanisms are active in rheumatoid arthritis, in which TGFβ level is elevated and correlates with osteoclast activity. Our findings identify a TGFβ/TNF-driven inflammatory osteoclastogenic program, and may lead to development of selective treatments for inflammatory osteolysis.

[1] Arthritis and Tissue Degeneration Program and David Z. Rosensweig Genomics Research Center, Hospital for Special Surgery, New York, New York, USA. [2] Department of Medicine, Weill Cornell Medical College, New York, NY, USA. [3] Department of Oncological Sciences, Icahn School of Medicine at Mount Sinai, New York, NY, USA. [4] Pathology and Laboratory Medicine, Weill Cornell Medical College, New York, NY, USA. [5] Research Institute, Hospital for Special Surgery, New York, NY, USA. [6] Graduate Program in Cell and Development Biology, Weill Cornell Graduate School of Medical Sciences, New York, NY, USA. [7] Present address: Xiangya Hospital, Central South University, Changsha, Hunan, China. [8] These authors contributed equally: Yuhan Xia, Kazuki Inoue. ✉email: zhaob@hss.edu

nflammatory bone loss by osteoclasts is a signature feature and severe consequence of many inflammatory disorders, such as rheumatoid arthritis (RA), periodontitis, and psoriatic arthritis. Bone destruction is also a major contributor to morbidity and disability in inflammatory arthritis patients[1–7]. Osteoclast cells derive from monocyte/macrophage lineage, and are the sole effective bone-resorbing cells. Osteoclasts play an important role not only in physiological bone development and remodeling, but also function actively to directly drive musculoskeletal tissue damage and accelerate the pathogenesis of diseases characterized by inflammatory osteolysis, such as RA[1–7]. Receptor activator of nuclear factor-kappa-B ligand (RANKL) is the only known physiological inducer of osteoclastogenesis. RANKL-induced osteoclastogenesis is important for maintaining normal bone remodeling and healthy bone mass in physiological/basal conditions. The mechanisms of RANKL-induced osteoclastogenesis have been extensively investigated and well defined. However, the mechanisms driving osteoclastogenesis under inflammatory conditions are complex and remain poorly understood, especially due to the fact that inflammatory cytokines such as tumor necrosis factor (TNF) display little to no direct ability to drive the differentiation of osteoclasts.

TNF is a crucial cytokine involved in immunity and inflammation, and plays a key role in driving chronic inflammation in a number of inflammatory and autoimmune diseases[8,9]. TNF stimulates bone erosion in many inflammatory diseases, such as in RA[3,5,10]. TNF blockade therapy has demonstrated the success in the inhibition of arthritic bone erosion[11–14]. However, how TNF drives its effects on bone resorption is a longstanding enigma given that TNF has only very modest direct osteoclastogenic effects on macrophages or osteoclast precursors, and needs to synergize with RANKL to promote osteoclast differentiation[6,10,15–19].

RANK receptor blockers and anti-RANKL neutralizing antibodies are recently available forms of treatment for excessive bone resorption, such as that occurring in osteoporosis, by inhibiting osteoclast formation. However, these treatments strongly inhibit osteoclast formation and could result in long-term side effects. Blocking RANKL signaling can lead to defective and dysregulated bone remodeling and bone repair, and risks of atypical femoral fractures and osteonecrosis of the jaw[20,21]. Discontinuation of denosumab, a RANKL inhibitor, is associated with rapidly rebound bone resorption and fracture risk[22]. TNF inhibitors have been utilized to treat RA-associated inflammation and joint erosion, but long-term usage has immunosuppressive side effects, such as common and opportunistic infections, and reactivation of latent tuberculosis[9]. Moreover, inflammatory bone erosion is resistant to standard anti-resorptive therapies[23–29]. This suggests that alternative molecular pathways for osteoclast formation and function may be active under inflammatory conditions, and such pathways need to be identified to develop anti-resorptive approaches that are effective in inflammatory diseases. Therefore, there is an important medical unmet need to discover these unknown pathways and mechanisms that mediate inflammatory osteoclastogenesis. These findings can provide attractive therapeutic developments to suppress inflammatory bone loss, whilst eliminating or minimizing negative effects on bone remodeling or immune response in physiological or disease settings. These pathways and mechanisms are currently poorly understood.

There is an abundant amount of transforming growth factor β (TGFβ) in bone tissue. TGFβ binds to its receptor composed of TGFβ receptor type II (Tgfbr2) and type I (Tgfbr1) and plays important roles in bone mass maintenance[30]. TGFβ is a multifunctional cytokine, which often interacts with other cytokines or growth factors to play diverse roles in different cells and settings[31,32]. The interaction between TGFβ and TNF is underappreciated. Despite a lot of effort having been put forth to

investigate the effect of TGFβ on RANKL-induced osteoclastogenesis, the results in the literature are controversial and often seemingly contradictory[33–38]. For example, TGFβ was reported to play a positive role in RANKL-induced osteoclastogenesis[33–35]. Other studies, however, found the inhibitory or dual (both inhibitory and stimulatory) effects of TGFβ on RANKL-induced osteoclastogenesis[36–38]. These studies do not have genetic evidence to support their results except that one group used the knockout mice of smad4[36], which does not represent TGFβ effect. In this study, we generated TGFβ receptor 2 conditional knockouts (KO) mice (Tgfbr2f/f;LysMCre) and show that deficiency of TGFβ signaling pathway in macrophages/osteoclast precursors does not affect RANKL-induced osteoclastogenesis in vitro and in vivo, or bone mass, providing clear evidence that TGFβ signaling is dispensable for RANKL-induced osteoclastogenesis under physiological conditions. In contrast, we found that TGFβ priming renders TNF sufficient to initiate a non-canonical, RANKL-independent pathway for osteoclast differentiation. This finding introduces a second molecular pathway for osteoclast generation distinct from the classical RANKL/RANK pathway. TGFβ shifts the pro-inflammatory action of TNF on macrophages to a highly efficient osteoclastogenic function by creating a favorable chromatin environment for osteoclastic gene expression. In mouse inflammatory models, blockade of TGFβ signaling protects mice from TNF-induced inflammatory bone erosion. As evidence of the potential human relevance of this pathway, TGFβ levels are significantly elevated in RA patients[39–44], and we found that TGFβ signaling pathway and osteoclastic gene expression are highly enriched and correlated in peripheral blood monocytes (PBMCs) isolated from RA patients. TGFβ and TNF-mediated osteoclastogenic mechanisms are active in RA, indicating a clinical relevance and presence of TGFβ and TNF-dependent inflammatory osteoclastogenic pathways in RA. This previously unrecognized non-canonical pathway for osteoclast generation mediated by TGFβ and TNF opens therapeutic avenues to treat inflammatory bone loss.

## Results
**TGFβ priming reprograms macrophage response to TNF towards osteoclastogenesis.** As inflammatory conditions may trigger as of yet unknown pathways for osteoclast generation, we sought to identify the key drivers of these alternative osteoclastogenic pathways. Since TGFβ level is elevated in RA[39–44], we considered whether TGFβ may fundamentally alter how macrophages respond to TNF, rendering it sufficient to drive osteoclast differentiation. To this end, we used a well-established and validated system with human nonproliferating CD14(+) peripheral blood monocyte (PBMC) derived macrophages, which are not only macrophages but are also circulating quiescent osteoclast precursors that migrate and attach to bone surface and differentiate into osteoclasts in response to RANKL[45,46]. The human system offers advantages of using cells directly relevant for human diseases and enabling future comparisons with patient cells. In this human system that mimics in vivo osteoclastogenesis, CD14(+) PBMCs are cultured with M-CSF for 3 days to induce macrophage differentiation (monocyte to macrophage stage, Supplementary Fig. 1), and then various treatments, such as TNFα (hereafter referred to as TNF) or RANKL, are applied to stimulate macrophages (macrophage response stage, Supplementary Fig. 1). As expected, TNF alone (hereafter referred to as non-priming) failed to induce osteoclast differentiation (1st lane, Fig. 1a). Treatment of macrophages with TGFβ1 (hereafter referred to as TGFβ) and TNF together induced TRAP(+) cells, but few giant multinuclear cells were observed (2nd lane, Fig. 1a). In striking contrast, TGFβ priming during the monocyte to

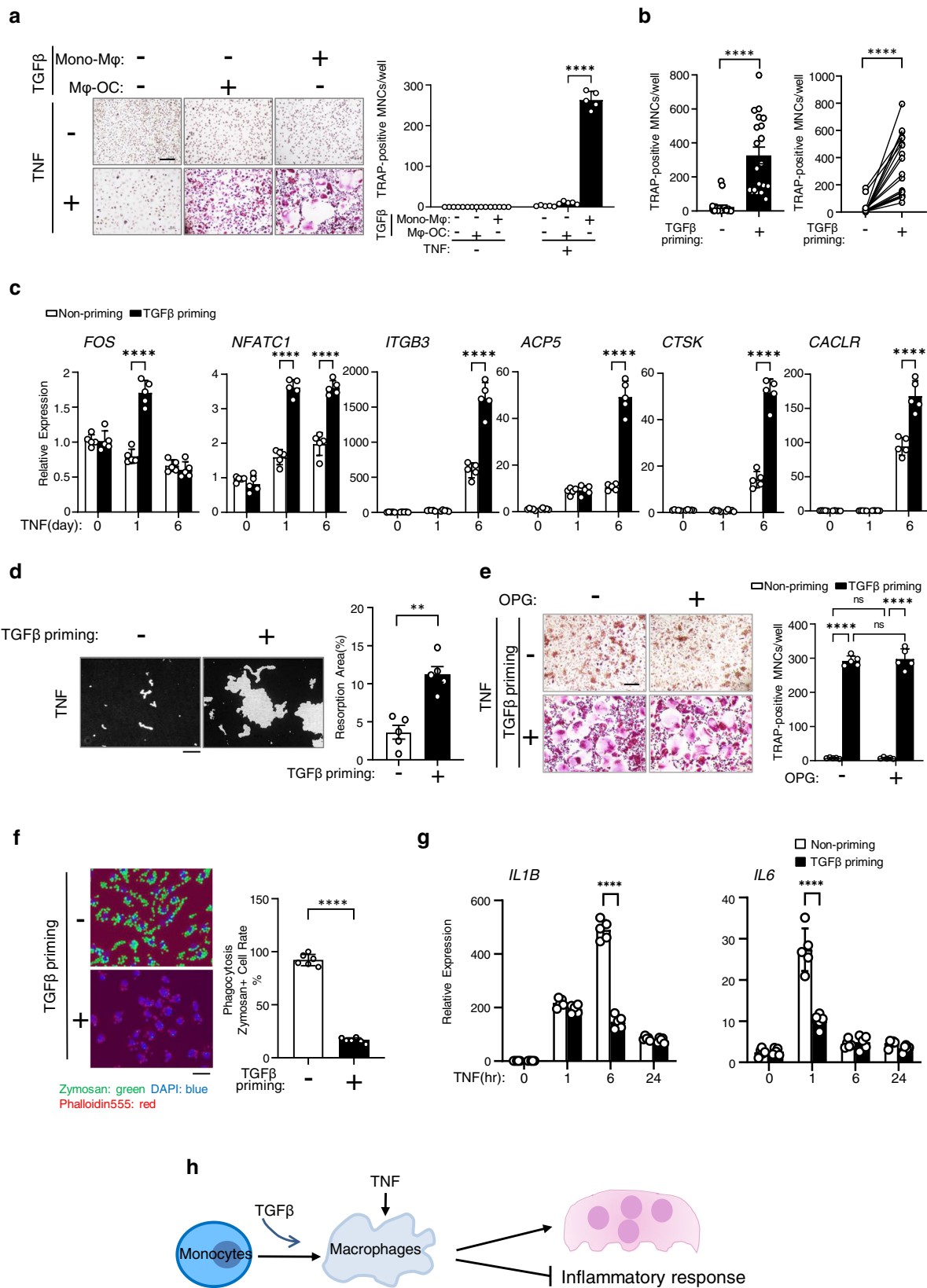

macrophage stage (TGFβ is removed before TNF stimulation; hereafter referred to as TGFβ priming) is able to effectively allow TNF to induce a number of giant multinuclear osteoclasts (3rd lane, Fig. 1a). TGFβ priming has a dose-dependent effect on TNF-mediated osteoclast differentiation (Supplementary Fig. 2a). Maintenance of TGFβ in the culture system throughout the

monocyte-macrophage-osteoclast stage also allows TNF to significantly induce a number of giant multinuclear osteoclasts (Supplementary Fig. 3). Notably, the TGFβ priming effect on TNF is highly consistent between donors (Fig. 1b). The expression of key osteoclastogenic transcription factors *FOS* (encoding c-Fos) and *NFATC1* (encoding NFATc1), and osteoclast marker

**Fig. 1 TGFβ priming switches the action of TNF to effectively drive osteoclastogenesis. a** Human osteoclast differentiation determined by TRAP staining (left) and the relative area of TRAP-positive multinuclear osteoclasts (MNCs, ≥3 nuclei/cell) per well (right) in the cell cultures using human CD14(+)-monocytes treated with or without TGFβ for 3 days, followed by TNF stimulation for 6 days in the presence or absence of TGFβ. Mono-Mφ: monocyte to macrophage stage. TRAP-positive cells: red. (n = 5/group). **b** Relative area of TRAP-positive-MNCs per well in human TNF-induced osteoclast differentiation with or without TGFβ priming (n = 20/group). **c** Quantitative real-time PCR (qPCR) analysis of mRNA expression of the indicated genes during osteoclastogenesis using human CD14(+)-monocytes treated with or without TGFβ priming for 3 days, followed by TNF stimulation for the indicated days (n = 5/group). **d** Von Kossa staining (left) and the resorption area (%) (right) of human osteoclast cultures induced by TNF with or without TGFβ priming (n = 5/group). Mineralized area: black; resorption area: white. **e** Human TNF-induced osteoclastogenesis using human CD14(+)-monocytes treated with or without TGFβ priming for 3 days, followed by TNF stimulation for 6 days in the presence or absence of recombinant OPG (100 ng/ml). Left: TRAP staining; Right: quantification of the relative area of TRAP-positive-MNCs/well (n = 5/group). **f** Phagocytosis of zymosan particles (left: zymosan staining; right: quantification of zymosan-containing cells) in the cell cultures using human CD14(+)-monocytes with or without TGFβ priming, followed by TNF stimulation for one day. Fluorescent zymosan particles: green; nuclei: blue; F-actin: red. (n = 6/group). **g** qPCR analysis of mRNA expression of *IL1B* and *IL6* using human CD14(+)-monocytes treated with or without TGFβ priming for 3 days followed by TNF stimulation for the indicated times (n = 5/group). **h** Schematic of TGFβ priming effects on TNF-mediated macrophage response. **a, c, e, g** **p < 0.01; ***p < 0.001; ****p < 0.0001; n.s. not statistically significant by two-way ANOVA with Bonferroni's multiple comparisons test. **b, d, f** **p < 0.01; ****p < 0.0001 by two-sided Student's t test. Error bars: **a, c, e, g**, Data are mean ± SD. **b, d, f** Data are mean ± SEM. Scale bars: **a, e, f** 200 μm; **d** 100 μm. Source data are provided as a Source Data file.

genes *ITGB3* (encoding β3 integrin), *ACP5* (encoding TRAP), *CTSK* (encoding cathepsin K) and *CALCR* (encoding Calcintonin receptor) was significantly enhanced by TGFβ priming (Fig. 1c). Furthermore, TNF-mediated osteoclasts generated after TGFβ priming are functional with effective mineral resorbing ability (Fig. 1d). The number of nuclei, the size, and the resorption activity of the TGFβ priming TNF-induced osteoclasts are comparable to those of RANKL-induced osteoclasts (Supplementary Fig. 4). Similar to the human culture system, we also found that TGFβ priming of murine bone marrow drastically induced osteoclast differentiation in response to TNF (Supplementary Fig. 2b, 5), indicating that TGFβ priming effect on TNF is conserved between mice and humans. Moreover, this effect is independent of RANKL, as evidenced by that an excessive amount of OPG (RANKL decoy receptor) does not impact TGFβ priming effect on TNF-mediated osteoclastogenesis in the human system (Fig. 1e) or mouse cell cultures (Supplementary Fig. 6a). In addition, TGFβ priming does not affect RANKL-induced osteoclastogenesis (Supplementary Fig. 6b). On the other hand, we examined whether TGFβ priming affects macrophage function. As shown in Fig. 1f, PBMC-derived macrophages phagocytized zymosan particles as expected. However, TGFβ priming almost completely abolished the phagocytic characteristics of macrophages (Fig. 1f). TGFβ priming also strongly suppressed the expression of cytokines IL-1B and IL-6 in macrophages stimulated with TNF (Fig. 1g). These results collectively indicate that TGFβ priming suppresses the inflammatory action of TNF and reprograms macrophage response towards osteoclastogenesis (Fig. 1h).

**Lack of TGFβ signaling prevents TNF, but not RANKL-induced osteoclastogenesis and bone resorption.** We next generated TGFβ receptor 2 conditional knockout (KO) mice, in which TGFβ receptor 2 is specifically deleted in myeloid lineage macrophages/osteoclast precursors by crossing *Tgfbr2^flox/flox* mice with *LysMcre* mice (*Tgfbr2^f/f*;*LysMCre*; hereafter referred to as *Tgfbr2^ΔM*, Supplementary Fig. 7a). Sex-matched *LysMcre+* littermates served as wild-type (WT) controls (hereafter referred to as WT). There is a basal level of TGFβ signaling in the physiological condition in WT mice, reflected by TGFβ target gene expression (Supplementary Fig. 7b), which is thought to correspond to a TGFβ priming condition in vivo. The macrophages in *Tgfbr2^ΔM* mice lack basal TGFβ signaling because of TGFβ receptor 2 deletion (Supplementary Fig. 7), mimicking the TGFβ non-priming condition in vivo. We wished to take advantage of these mouse models to provide a proof of concept for TGFβ priming effect on osteoclast differentiation and bone resorption

activity in vivo. We first examined RANKL-induced osteoclastogenesis using bone marrow-derived macrophages (BMMs). We found that TGFβ signaling deficiency in *Tgfbr2^ΔM* BMMs did not affect osteoclast differentiation induced by RANKL (Fig. 2a). Consistent with this in vitro osteoclast differentiation induced by RANKL, *Tgfbr2^ΔM* mice did not exhibit significant defects in the bone phenotype of femoral trabecular and cortical bones and vertebral trabecular bones compared with the WT control littermates (Fig. 2b, c, Supplementary Fig. 8a, b). These data indicate that TGFβ signaling plays a dispensable role in the osteoclast differentiation process in the physiological setting. We further considered whether the lack of TGFβ receptor 2 impacts osteoclastogenesis and bone resorption in response to TNF. Compared to the full osteoclastogenesis primed by TGFβ in WT control cells, TNF failed to induce osteoclast differentiation in TGFβ priming condition in *Tgfbr2^ΔM* cell cultures supported by few osteoclast formations and weak osteoclast marker gene expression (Fig. 2d, e). To test the effect of TGFβ on TNF response in vivo, we employed a well-established mouse calvarial osteolysis model through inflammatory response to TNF[47,48]. PBS injection was used as the negative control and no resorptive pit formation was observed on calvarial bone surfaces, and *Tgfbr2* deficiency in myeloid lineage did not affect basal osteoclastogenesis (Fig. 2f, g, h). *Tgfbr2* deficiency in the *Tgfbr2^ΔM* mice significantly prevented TNF-induced erosions, as evidenced through μCT analysis of resorption pit formation on the calvarial bone surface (Fig. 2f), and substantially decreased osteoclast formation on the calvarial bone histological slices compared to the WT control (Fig. 2g, h). We further examined bone phenotypes of *Tgfbr2^ΔM* mice and the WT control mice using a long-term TNF-induced inflammatory bone resorption model. In this model, recombinant TNF was injected into the calvarial periosteum daily for 14 days to mimic a chronic TNF-induced inflammatory condition in vivo. Compared to the PBS injection groups as the control, long-term TNF treatment induces not only bone resorption on the calvarial bones (Supplementary Fig. 9a), but also significant trabecular bone loss in femurs and vertebrae in the WT mice (Supplementary Fig. 9b, c). In striking contrast, the lack of TGFβ signaling protects against bone loss in calvarial, femoral and vertebral bones induced by TNF in the *Tgfbr2^ΔM* mice (Supplementary Fig. 9a–c). These results furthermore demonstrate that TGFβ signaling plays a crucial role in enhancing TNF-mediated inflammatory bone loss.

We next explored the role of TGFβ priming in an inflammatory arthritis model which mimics human pathology and would provide greater pathological relevance. We selected the K/BxN serum-induced arthritis model[49], which resembles RA and therefore has been widely used to study inflammation-

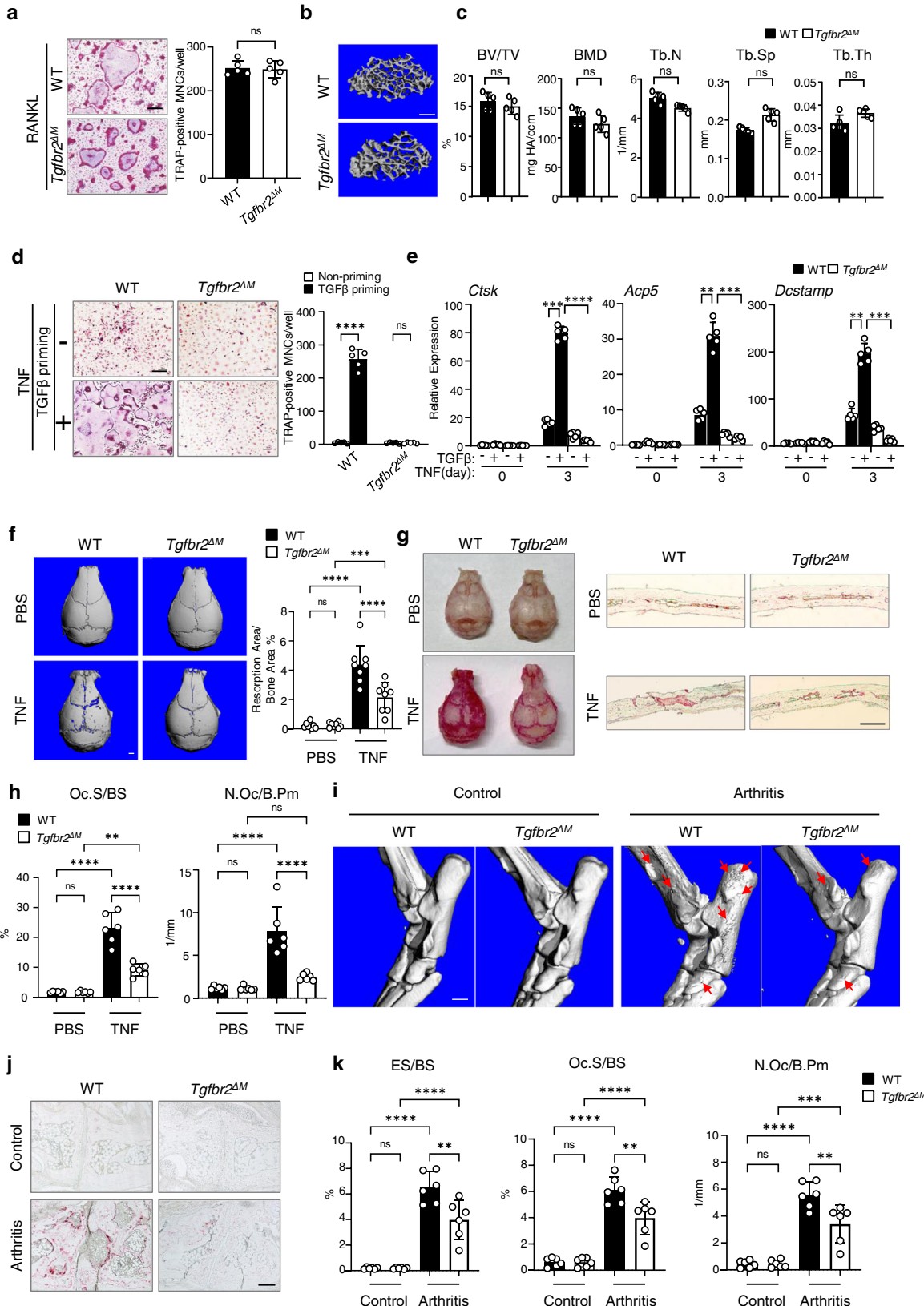

induced peri-articular bone erosion by inflammatory cytokines, including TNF. This model allows the investigation of bone resorption during the effector phase of inflammatory arthritis. Deficiency of basal TGFβ signaling (non-priming) in *Tgfbr2^{ΔM}* mice resulted in significant suppression of peri-articular bone erosion, and reduction of osteoclast numbers and surface in

resorption sites (Fig. 2i–k) of the tarsal joints in the K/BxN arthritis model in comparison to the WT control mice that mimic TGFβ priming condition. Joint swelling was sustained during the 14-day course of inflammatory arthritis, while suppressed bone erosion was observed in *Tgfbr2^{ΔM}* mice (Supplementary Fig. 10), suggesting that TGFβ signaling has no significant effect on

**Fig. 2 Loss of TGFβ signaling suppresses inflammatory osteoclastogenesis and bone resorption. a** Osteoclast differentiation using BMMs derived from WT and *Tgfbr2*[ΔM] mice was stimulated with RANKL for 3 days. Left: TRAP staining. Right: relative area of TRAP-positive-MNCs (≥3 nuclei/cell) per well. TRAP-positive cells: red. (*n* = 5/group). **b, c** μCT images (**b**) and bone morphometric analysis (**c**) of trabecular bone of the distal femurs isolated from the indicated 12-week-old-male littermate mice (*n* = 5/group). **d, e** Osteoclast differentiation determined by TRAP staining (**d**, left) and the relative area of TRAP-positive MNCs per well (**d**, right), and qPCR analysis of mRNA expression of *Ctsk, Acp5,* and *Dcstamp* (**e**) in the cell cultures, in which the bone marrow of WT and *Tgfbr2*[ΔM] mice was primed with or without TGFβ for 4 days, followed by TNF stimulation for 3 days. TRAP-positive cells: red. **d, e**: *n* = 5/group. **f–h** μCT images (**f**, left), the quantification of the resorption area (**f**, right), TRAP staining of bone surface (**g**, left) and histological sections (**g**, right), and histomorphometric analysis of the slices of the calvarial bones obtained from 12-week-old-male WT and *Tgfbr2*[ΔM] mice after PBS or TNF injection to the calvarial periosteum daily for 5 days (*n* = 8/group in **f**, *n* = 6/group in **h**). **i–k** μCT images of the surface of tarsal joints (red arrows: bone erosion) (**i**), TRAP staining (**j**) and histomorphometric analysis (**k**) of the tarsal joint sections obtained from the indicated 12-week-old female mice with PBS injection as the control or the littermate female mice that developed K/BxN serum-induced arthritis (*n* = 6/group). BV/TV, bone volume per tissue volume; BMD, bone mineral density; Tb.N, trabecular number; Tb.Sp, trabecular separation; Tb.Th, trabecular thickness; ES/BS, erosion surface/bone surface; Oc.S/BS, osteoclast surface/bone surface; N.Oc/B.Pm, number of osteoclasts per bone perimeter. **a, c** ns: not statistically significant by two-sided Student's *t* test. **d, e, f, h, k** \*\**p* < 0.01; \*\*\**p* < 0.001; \*\*\*\**p* < 0.0001; ns: not statistically significant by two-way ANOVA with Bonferroni's multiple comparisons test. Error bars: **a, c–f, h, k** Data are mean ± SD. Scale bars: **a, d, j** 200 μm; **b, g** 100 μm; **f, i** 1.0 mm. Source data are provided as a Source Data file.

inflammation, but prominently affects osteoclastogenesis and bone erosion in this model.

Taken together, our results suggest that endogenous TGFβ signaling does not affect RANKL-induced osteoclast formation, but plays a crucial role in promoting TNF-mediated osteoclastogenesis and exacerbating inflammatory bone resorption.

**TGFβ priming suppresses the expression of IFN-stimulated genes, meanwhile switching TNF to elicit osteoclastogenic gene induction.** We next investigated the mechanisms by which TGFβ priming reprograms macrophages response to TNF towards osteoclastogenesis. To address this, we performed gene expression profiling using high-throughput sequencing of RNAs (RNA-seq) from the human culture system to identify transcriptomic changes by TGFβ priming. Consistent with prior literature[50,51], pathway analysis showed highly up-regulated genes involved in IFN, particularly type-I IFN signaling pathway, and inflammatory response by TNF stimulation of macrophages (non-priming condition, Fig. 3a). TGFβ priming, however, completely enriched different gene sets induced by TNF that appear to be required for cell differentiation, such as ribosomal proteins and osteoclast differentiation (Osteoclast differentiation and Focal Adhesion) (Fig. 3b). RNA-seq–based expression heatmaps revealed that many IFN-stimulated genes (ISGs), including type-I IFN response genes (Fig. 3c) and chemokine genes (Fig. 3d), were exclusively induced by TNF after 24 h of stimulation. In striking contrast, under the TGFβ priming condition, TNF did not induce these inflammatory genes, but instead activated many classic osteoclastic genes (hereafter referred to as OC genes), including the early-stage osteoclastogenic regulators, such as *FOS, NFATC1,* and *MITF* (d1, Fig. 3e), and a set of late-stage osteoclast marker genes, such as *ACP5, CTSK, ITGB3, OSCAR* and *CALCR* (d6, Fig. 3e). Gene set enrichment analysis (GSEA) also confirmed that type-I IFN response genes and chemokine genes comprised the top significant gene set enrichment by TNF stimulation of macrophages, whereas TNF with TGFβ priming showed osteoclastic gene set enrichment (Fig. 3f–h). These data support our findings that TGFβ priming enabled TNF-induced osteoclastogenesis (Fig.1). We further confirmed the expression of type-I IFN response genes, such as *IFIT1, IFIT2, MX1,* and *STAT1* (Fig. 3i), and chemokine genes, such as *CCL5, CXCL9,* and *CXCL10* (Fig. 3j), which was induced by TNF at 24 h but almost completely abolished by TGFβ priming. These transcriptomic findings indicate that TGFβ priming prominently plays a selective role in negative regulation of ISG genes but positive regulation of osteoclastic genes in macrophages stimulated by TNF.

**TGFβ priming remodels chromatin accessibility and histone marks to reprogram gene transcription towards osteoclastogenesis.** The selective TGFβ priming effect on gene expression led us to further investigate the epigenetic basis of how TGFβ fundamentally alters the macrophage response to TNF to enable its osteoclastogenic actions. Chromatin accessibility and histone modifications often change dynamically in response to environmental cues and play fundamental roles in the epigenetic regulation of gene expression. To address this, we first performed ATAC-seq, and found that TNF significantly induced differentially accessible ATAC-seq peaks in the TGFβ non-priming and priming conditions (Fig. 4a), indicating that TGFβ priming largely altered chromatin accessibility. Pathway analysis of the genes associated with these differentially accessible peaks in each condition showed that TGFβ priming shifted TNF-increased chromatin accessibility from genes enriched in type-I IFN response and chemokine pathways to the genes involved in TGFβ signaling and osteoclast differentiation pathway (Fig. 4b). These chromatin accessibility results are consistent with the transcriptomic effects of TGFβ priming (Fig. 3). We then conducted an integrative ATAC-seq and RNA-seq analysis to elucidate the chromatin accessibility associated with the differentially expressed genes (DEGs) arising in response to TNF with or without TGFβ priming. We term the DEGs, whose expression was enhanced by TNF in the TGFβ priming condition, as TGFβ priming genes, including TGFβ pathway genes and OC genes. The DEGs enhanced by TNF in the non-priming condition were referred to as non-priming genes, including ISGs. We found that 60% of non-priming gene loci (745 genes) associated with 1975 peaks showed increased chromatin accessibility in response to TNF stimulation (Fig. 4c, d), whereas TGFβ priming significantly reduced the chromatin accessibility at these loci at both baseline and/or after TNF stimulation, such as the loci of MX1, MX2, IFIT2, IFIT3, CXCL9, CXCL10 and CXCL11 (Fig. 4c, d, m). In contrast, TGFβ priming increased chromatin accessibility at the TGFβ priming gene loci regardless of TNF treatment, such as the loci of NFATC1, ACP5, ITGB3, and CALCR (Fig. 4c, d, n, and Supplementary Fig. 11). We further assessed ISG and OC loci, and found similar changes in chromatin accessibility as those seen at non-priming and priming genes, respectively (Fig. 4e). The TGFβ-induced chromatin closing was observed at all ISG loci. Notably, TNF alone does not change the chromatin accessibility at OC gene loci (Fig. 4e, right panel), which is consistent with the results that TNF alone is not able to effectively induce OC gene expression (Fig. 3e). Most OC gene loci (81%) were markedly opened by TGFβ priming before TNF stimulation despite the that no gene expression was observed at baseline

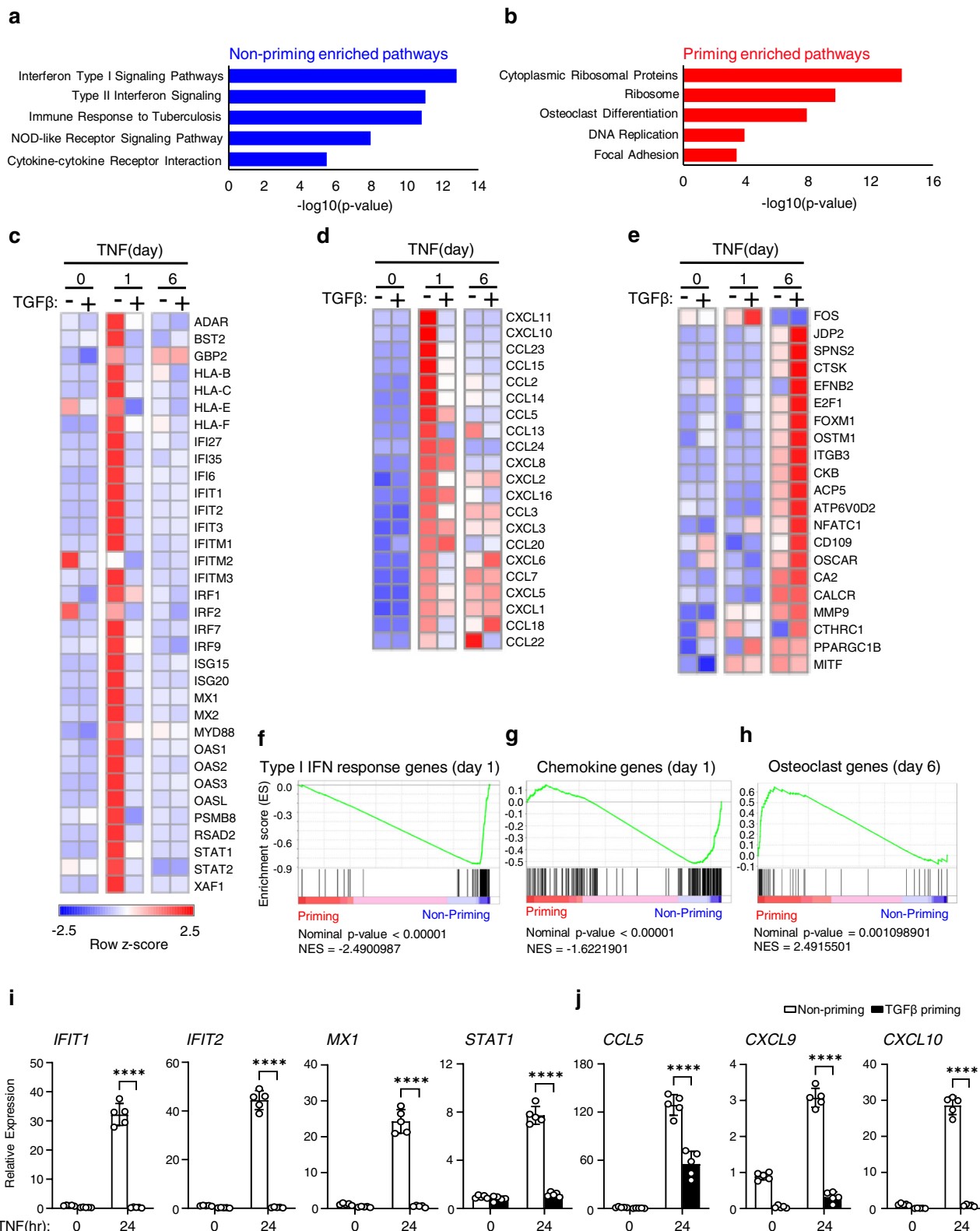

**Fig.** (a) Non-priming enriched pathways. (b) Priming enriched pathways. (c-e) Heatmaps of gene expression with TNF(day) 0, 1, 6 and TGFβ -/+. (f) Type I IFN response genes (day 1); Nominal p-value < 0.00001; NES = -2.4900987. (g) Chemokine genes (day 1); Nominal p-value < 0.00001; NES = -1.6221901. (h) Osteoclast genes (day 6); Nominal p-value = 0.001098901; NES = 2.4915501. (i-j) Relative expression of IFIT1, IFIT2, MX1, STAT1, CCL5, CXCL9, CXCL10 at TNF(hr) 0 and 24. Non-priming vs TGFβ priming. **** indicated.

without TNF stimulation. These findings indicate that TGFβ priming prepared and enabled these gene loci to become accessible for gene induction by subsequent stimulatory signals. One exception was the FOS locus, whose accessibility was not increased by TGFβ priming at baseline but enhanced after TNF stimulation (Fig. 4n). With TGFβ priming, TNF stimulation further increased chromatin accessibility at some OC gene loci

(Fig. 4e, right panel), such as the CALCR locus. We further performed FAIRE (formaldehyde-assisted isolation of regulatory element)-qPCR assay to quantify compaction and accessibility of chromatin at the ISG and OC gene loci during chromatin remodeling at the late stage (d6) of osteoclastogenesis. The results show that TGFβ priming continuously drastically enhances the chromatin accessibility at the loci of OC genes, such as NFATC1,

**Fig. 3 TGFβ priming reprograms the action of TNF on the transcriptome in macrophages from inflammatory towards osteoclastic gene expression.**
**a–h** RNA-seq analysis of the mRNAs from human CD14(+) monocytes treated with or without TGFβ priming for 3 days, followed by TNF stimulation for 1 or 6 days. $n = 3$ biological replicates for each condition. **a** Pathway analysis of TNF-inducible genes at 24 h in the non-priming condition. **b** Pathway analysis of TNF-inducible genes at day 6 in the TGFβ priming condition. **c–e** Heatmaps of TNF-induced type-I IFN response genes (**c**), chemokine genes (**d**), and osteoclast genes (**e**) regulated by non-priming and TGFβ priming at the indicated times. Row $z$ scores of CPMs were shown in the heatmaps. **f–h** Gene set enrichment analysis of TNF-inducible type-I IFN response genes (**f**), chemokine genes (**g**), and osteoclast genes (**h**) regulated by non-priming and TGFβ priming ranked by NES. **i–j** qPCR analysis of mRNA expression of *IFIT1*, *IFIT2*, *MX1*, *STAT1*, *CCL5*, *CXCL9* and *CXCL10* using human CD14(+) monocytes treated with or without TGFβ priming for 3 days, followed by TNF for 1 day ($n = 5$/group). **i**, **j** ***$p < 0.001$ by two-way ANOVA with Bonferroni's multiple comparisons test. Error bars: **i**, **j** Data are mean ± SD. Source data are provided as a Source Data file.

ITGB3, ACP5, CALCR, and MYBL2, at day 6 of TNF stimulation of macrophages (Supplementary Fig. 12a). Fos locus is an exception in that its chromatin became closed on day 6 regardless of TGFβ priming or not (Supplementary Fig. 12a). In contrast, most ISG gene loci, such as MX1, MX2, IFIT2, IFIT3, CXCL9, CXCL10, CXCL11, IFNB1, and IRF1, at day 6 of TNF treatment turn into a closed state, which is not affected or further suppressed by TGFβ priming (Supplementary Fig. 12b). These chromatin remodeling changes at a late stage of TNF stimulation of macrophages are consistent with the transcriptomic results of OC genes and ISG genes shown in Fig. 3c–e, in which ISG gene expression (Fig. 3c, d) is almost completely suppressed at day 6 of TNF stimulation, while the expression of OC genes, except for FOS, is highly induced at day 6 of TNF stimulation in the TGFβ priming condition (Fig. 3e). We next performed Cut&Run-seq for the H3K4me3 histone mark associated with transcriptional activation and the H3K27me3 histone mark associated with transcriptional inhibition. H3K4me3 signals were identified around the transcriptional start site (TSS) in most non-priming genes, including ISGs, at baseline and after TNF stimulation (Fig. 4f, g, m). These H3K4me3 signals were drastically attenuated by TGFβ priming (Fig. 4f, g, m). Compared with the non-priming genes, both the basal and TNF-stimulated levels of H3K4me3 mark at the priming genes were low, but TGFβ priming strikingly increased H3K4me3 signals at these gene loci, including OC genes NFATC1, FOS, ACP5, ITGB3 and CALCR (Fig. 4f, g, n, and Supplementary Fig. 11). H3K27me3 mark was nearly undetectable at ISG loci. In contrast, most OC genes (69%), including NFATC1, ITGB3, and CALCR, exhibited a basal H3K27me3 mark, which was drastically attenuated not by TNF treatment, but by TGFβ priming (Fig. 4h, i, n, and Supplementary Fig. 11). We further performed Cut&Run-seq for the H3K27ac histone mark, which is often enriched at active regulatory elements and associated with active gene transcription[52,53]. Indeed, H3K27ac signals were detected around the ATAC peaks. At non-priming gene loci, including ISGs, H3K27ac signals were present at baseline and increased after TNF stimulation (Fig. 4j–m). These H3K27ac signals were drastically attenuated by TGFβ priming (Fig. 4j–m). On the contrary, both basal and TNF-stimulated levels of H3K27ac signals were low at priming gene loci, such as OC genes, but TGFβ priming strikingly elevated these H3K27ac signals with even higher levels with TNF stimulation (Fig. 4j–n, and Supplementary Fig. 11). These H3K27ac signal changes are consistent with the H3K4me3 mark and transcriptomic changes of ISGs and OC genes.

In summary, most OC gene loci are largely closed with low levels of H3K4me3/H3K27ac and high H3K27me3 marks in macrophages, pointing to an inactive transcriptional state of these genes in macrophages. Although TNF signaling alone effectively increases chromatin accessibility at ISG loci, it fails to open the loci of OC genes. TGFβ priming, however, switches the chromatin state of ISGs and OC genes to the opposite direction, which initiates this to occur at the basal level. Chromatin becomes closed and H3K4me3/H3K27ac signal is reduced at ISG loci by

TGFβ priming. On the other hand, TGFβ priming strongly increases chromatin accessibility and H3K4me3/H3K27ac signal, while reducing the suppressive H3K27me3 mark at OC gene loci. This powerful TGFβ priming-induced chromatin remodeling at different gene sets provides an essential prerequisite and foundation for reprogramming the gene expression program, leading to an effective induction of OC gene transcription by TNF in macrophages instead of ISGs. Thus, the differential expression of ISGs and OC genes is tightly associated with chromatin remodeling induced by TGFβ priming.

**TGFβ priming suppresses the TNF-induced IRF1-IFNβ axis.** We next asked what transcriptional factors and pathways might coordinate with chromatin remodeling to regulate gene expression primed by TGFβ. De novo motif analysis of DNA sequences enriched under the ATAC-seq peaks associated with non-priming genes and TGFβ priming genes showed the top significantly enriched transcription factor binding sites for the non-priming genes include ISRE (IRF), AP-1, NF-κB, and IRF1 (Fig. 5a left panel). These transcription factors drive the expression of ISGs and inflammatory genes in response to TNF stimulation (Fig. 3 [50,51]). TGFβ priming enriched distinct transcription factor binding sites for the priming genes, which include FOS, E2F, SMAD2/3/4, MYBL2, and MITF (Fig. 5a right panel). SMAD2/3/4 are key transcription factors for TGFβ-induced Smad pathway[54,55]. FOS, E2F, and MITF are important osteoclastogenic regulators[46,56]. Hence, in parallel to chromatin remodeling, TGFβ priming alters transcription factor enrichment at different gene loci.

Since TNF induces ISGs via endogenous IFNβ production[50], we next examined the effect of TGFβ priming on IFNβ expression. Similar to the effects seen at ISG loci, TGFβ priming substantially reduced chromatin accessibility and the levels of H3K4me3/H3K27ac at the *IFNB1* promoter and 3′ downstream regulatory regions (Fig. 5b). As a consequence, TGFβ priming almost completely abolished TNF-induced IFNB1 gene expression (Fig. 5c) and protein production (Fig. 5d). Lack of TGFβ signaling, on the other hand, promoted TNF-induced IFNβ production (Fig. 5e), which provided complementary evidence for the suppressive effect of TGFβ priming on IFNβ expression. TNF induces IFNβ expression mainly via IRF1[50]. We found that the chromatin accessibility and H3K4me3/H3K27ac signal at the promoter and enhancer regions of IRF1 locus were significantly decreased by TGFβ priming (Fig. 5f), which is aligned with the changes at the IFNB1 locus. As expected, TGFβ priming drastically suppressed TNF-induced mRNA (Fig. 5g) and protein expression (Fig. 5h) of IRF1. IFNβ is a well-established inhibitor of osteoclast differentiation[48,57]. Thus, the TGFβ priming-induced IFNβ reduction is considered to contribute to enhanced osteoclastogenesis. On the other hand, in contrast to the strong IRF1 induction by TNF, RANKL does not induce IRF1 expression (Fig. 5i), which may explain the phenotype that IRF1 deficiency does not affect RANKL-induced osteoclastogenesis (Supplementary Fig. 13). However, lack of IRF1 dramatically enhanced TNF-

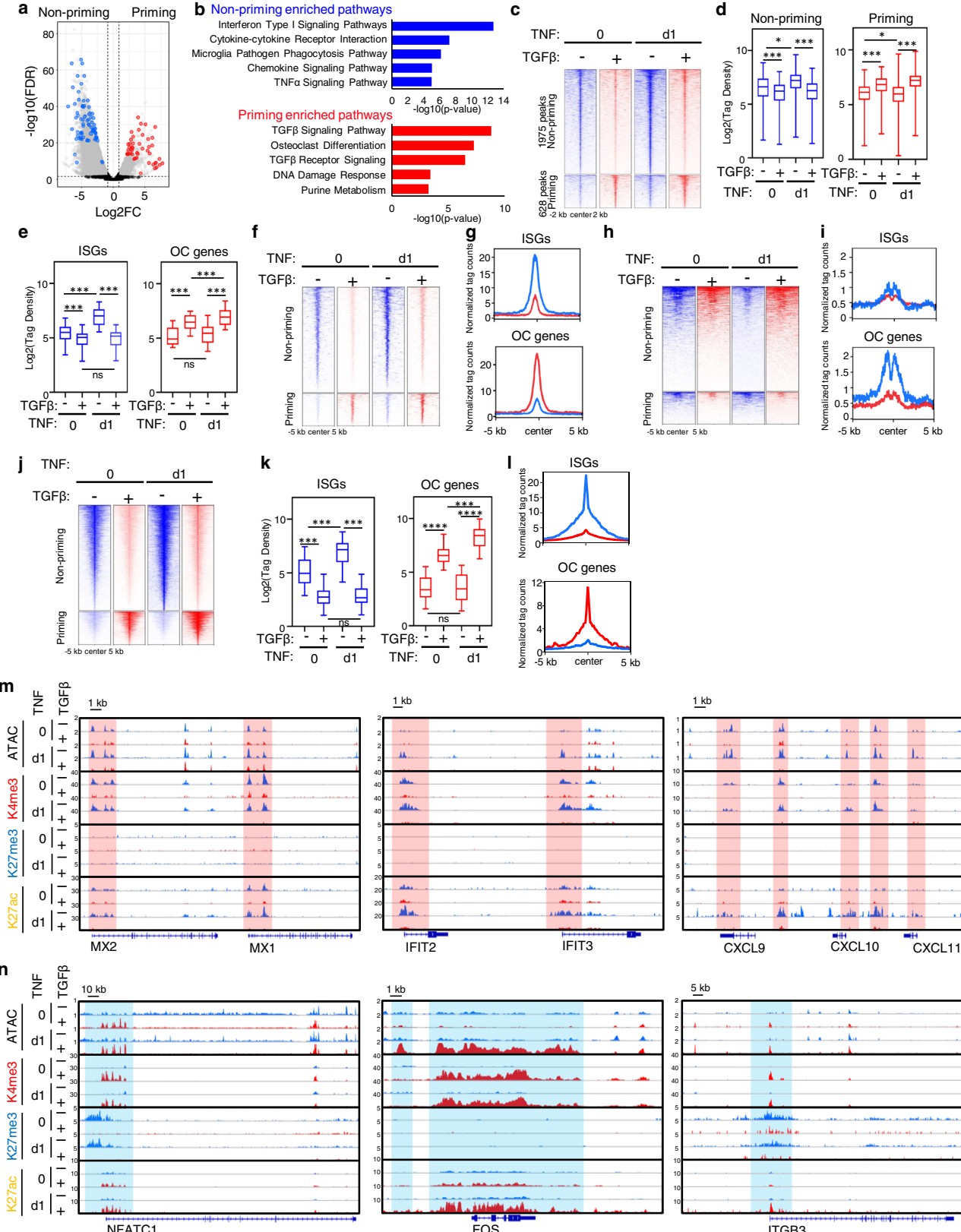

mediated osteoclast differentiation (Fig. 5j) and osteoclastic gene expression (Fig. 5k) in the TGFβ priming condition. TGFβ priming-suppressed ISG induction, such as that of *Ifit1, Ifit2, Mx1, and Stat1*, was further significantly suppressed by IRF1 deficiency (Fig. 5l). These results collectively demonstrate that TGFβ priming suppresses the TNF-induced IRF1-IFNβ axis and

thereby contributes to ISG gene inhibition and osteoclastic promotion.

**TGFβ priming promotes IRF8 protein degradation.** IRF8 is a potent inhibitory transcription factor of osteoclast

**Fig. 4 TGFβ priming regulates chromatin accessibility and histone modification to suppress the inflammatory action of TNF and facilitate osteoclastogenesis.** Human CD14(+)-monocytes were treated with or without TGFβ priming for 3 days, followed by TNF stimulation for 0 or 1 day. **a** Volcano plot of ATAC-seq analysis of TNF-induced differentially accessible peaks at day 1 (gray dots) with significant (FDR < 0.01) and greater than twofold changes between non-priming and TGFβ-priming conditions. Data are from two biological replicates. Blue dots: peaks associated with ISGs. Red dots: peaks associated with osteoclast and TGFβ signaling genes. **b** Pathway analysis of genes associated with the significantly differentially accessible peaks identified in **a**. **c**, **d** Normalized ATAC-seq tag-density (heatmap in **c**) and tag counts (boxplots in **d**) of differentially accessible peaks associated with non-priming or TGFβ-priming genes. **e** Boxplots showing normalized ATAC-seq tag counts of differentially accessible peaks associated with ISGs or osteoclast genes. **f** Heatmap of normalized H3K4me3 Cut&Run-seq tag-density of the differentially accessible peaks associated with non-priming or TGFβ-priming genes. **g** Average tag density profile of normalized H3K4me3 Cut&Run-seq peaks associated with ISGs or osteoclast genes. Blue: Non-priming; Red: TGFβ-priming. **h** Heatmap of normalized H3K27me3 Cut&Run-seq tag-density of the differentially accessible peaks associated with non-priming or TGFβ-priming genes. **i** Average tag-density profile of normalized H3K27me3 Cut&Run-seq peaks associated with ISGs or osteoclast genes. Blue: Non-priming; Red: TGFβ-priming. **j** Heatmap of normalized H3K27ac Cut&Run-seq tag-density of the differentially accessible peaks associated with non-priming or TGFβ-priming genes. **k** Boxplots showing normalized H3K27ac Cut&Run-seq counts of peaks associated with ISGs or osteoclast genes. **l** Average tag-density profile of normalized H3K27ac Cut&Run-seq peaks associated with ISGs or osteoclast genes. Blue: Non-priming; Red: TGFβ-priming. **m**, **n** Representative IGV tracks displaying normalized tag-density profiles for ATAC-seq, H3K4me3, H3K27me3, and H3K27ac Cut&Run-seq signals at ISG (**m**) and osteoclast gene loci (**n**). **d**, **e**, **k**, **m**, **n** Data are representative of 2 biological replicates. Data are presented as normalized tag density ±2 kb (**c**) or 5 kb (**f**, **h**, **j**) around peak centers. **d**, **e**, **k** *$p < 0.05$; ***$p < 0.001$, ****$p < 0.0001$; ns, not statistically significant by two-way ANOVA with Bonferroni's multiple comparisons test. Boxes represent data within the 25th to 75th percentiles. Whiskers depict the range of min to max. Horizontal lines within boxes represent median values. Source data are provided as a Source Data file.

differentiation[57,58]. Downregulation of IRF8 expression is a prerequisite for osteoclastogenesis. TNF alone did not significantly decrease IRF8 protein level, but TGFβ priming enabled TNF to quickly and substantially downregulate IRF8 protein (Fig. 6a). The diminishment of IRF8 protein by TGFβ priming eliminated an important negative regulator of osteoclastogenesis, facilitating the differentiation process. Interestingly, TGFβ priming did not affect the decrease of IRF8 mRNA levels by TNF (Fig. 6b), which is different from the TGFβ priming effect on most other genes at the transcriptional level. This suggests a unique mechanism by which TGFβ priming regulates IRF8 protein expression. We used MG132 to block proteasome activity and observed that the diminished IRF8 protein level in the TGFβ priming condition was fully restored to a level similar to that without priming (Fig. 6c). These results indicate that TGFβ priming decreases the protein stability of IRF8 and promotes its degradation, which is also true when protein synthesis is blocked (Fig. 6d). In line with this, the ubiquitination of IRF8 was significantly increased in TGFβ priming conditions when proteasome activity was inhibited (Fig. 6e), revealing that IRF8 protein degradation is largely promoted by TGFβ priming.

**TGFβ priming enables TNF to induce non-canonical osteoclastogenic regulators distinct from those induced by RANKL.** As TGFβ priming enables TNF to effectively induce osteoclastogenesis, we wondered whether this TGFβ/TNF and the canonical RANKL pathways for osteoclastogenesis induce a convergent or divergent set of response genes. As shown in Fig. 7a, there are a number of overlapping genes that can be induced by both TGFβ priming/TNF and RANKL. Further pathway analysis of these overlapped genes revealed significantly enriched pathways, including Osteoclast differentiation and Focal adhesion pathway, which are classic RANKL-induced osteoclastogenic pathways (Fig. 7a, bottom panel). Consistent with this, common osteoclastogenic transcription factors, such as *PU.1, NFATC1, FOS, PRDM1, E2F1, and MITF*, and osteoclast marker genes, such as *CTSK, ACP5, ITGB3, and CALCR*, were found in the overlapped gene sets (non-DEGs) induced by either TGFβ priming/TNF or RANKL stimulation (Fig. 7b). We further conducted the de novo motif analysis of sequences enriched under the ATAC-seq peaks associated with OC genes induced by TGFβ priming/TNF, and found the enrichment of the binding sites for common osteoclastogenic transcription factors E2F1, PRDM1, NFATC1, FOS, PU.1 and

MITF (Fig. 7c). Different from these common/classic OC transcription factors, we found an exceptionally enriched transcription factor binding site for MYBL2 (encoding B-Myb) (Fig. 7c), which is a highly expressed DEG induced by TNF in the TGFβ priming condition, but not by RANKL (Fig. 7b). B-Myb is a transcription factor that regulates hematopoietic progenitor cell development[59]. The function of B-Myb in osteoclast differentiation is unclear. The chromatin accessibility at MYBL2 locus is very low at baseline with high suppressive H3K27me3 mark. TGFβ priming significantly opened the chromatin at promoter and enhancer regions of MYBL2, decreased H3K27me3 mark, and enhanced H3K4me3/H3K27ac signals (Fig. 7d). Thus, MYBL2 is a typical TGFβ priming gene. Consistent with these chromatin remodeling and histone mark changes, MYBL2 expression at the basal level was low, and TGFβ priming markedly elevated TNF-induced MYBL2 expression (Fig. 7e, f). In order to examine the function of B-Myb in osteoclast differentiation, we knocked down its expression (Fig. 7g), and found that the TNF-induced osteoclast differentiation in the TGFβ priming condition was significantly suppressed by MYBL2 deficiency, evidenced by the reduced osteoclast formation (Fig. 7h) and attenuated osteoclast marker gene expression (Fig. 7i). In contrast, RANKL did not induce MYBL2 expression (Fig. 7b, j), and accordingly, RANKL-induced osteoclastogenesis was not affected by MYBL2 loss (Fig. 7k). Consistent with our finding that MYBL2 is a TGFβ priming gene, MYBL2 deficiency did not impact TNF-induced expression of ISGs, such as *IFIT1, IFIT2* and *MX1* (Fig. 7l). The expression of B-Myb was identified in osteoclasts in the TNF-induced supracalvarial osteolysis model in vivo (Fig. 7m). We further generated myeloid specific *Mybl2* conditional KO mice (*Mybl2^{f/f};LysMCre*; hereafter referred to as *Mybl2^{ΔM}*) and examined the effects of myeloid cell-specific *Mybl2* deletion on TNF-induced inflammatory osteoclastogenesis and bone resorption in vivo. Compared to the WT control mice, *Mybl2* deficiency significantly suppressed TNF-induced calvarial bone erosions (Fig. 7n), and markedly decreased osteoclast formation in the calvarial bones from the *Mybl2^{ΔM}* mice (Fig. 7o, p, TNF group). Myeloid cell-specific *Mybl2* deletion did not affect osteoclast formation under physiological condition (Fig. 7o, p, PBS group). Therefore, B-Myb is a previously unrecognized osteoclastogenic regulator that specifically promotes TNF-mediated osteoclastogenesis in the TGFβ priming condition, but not RANKL-induced osteoclastogenesis. Despite that

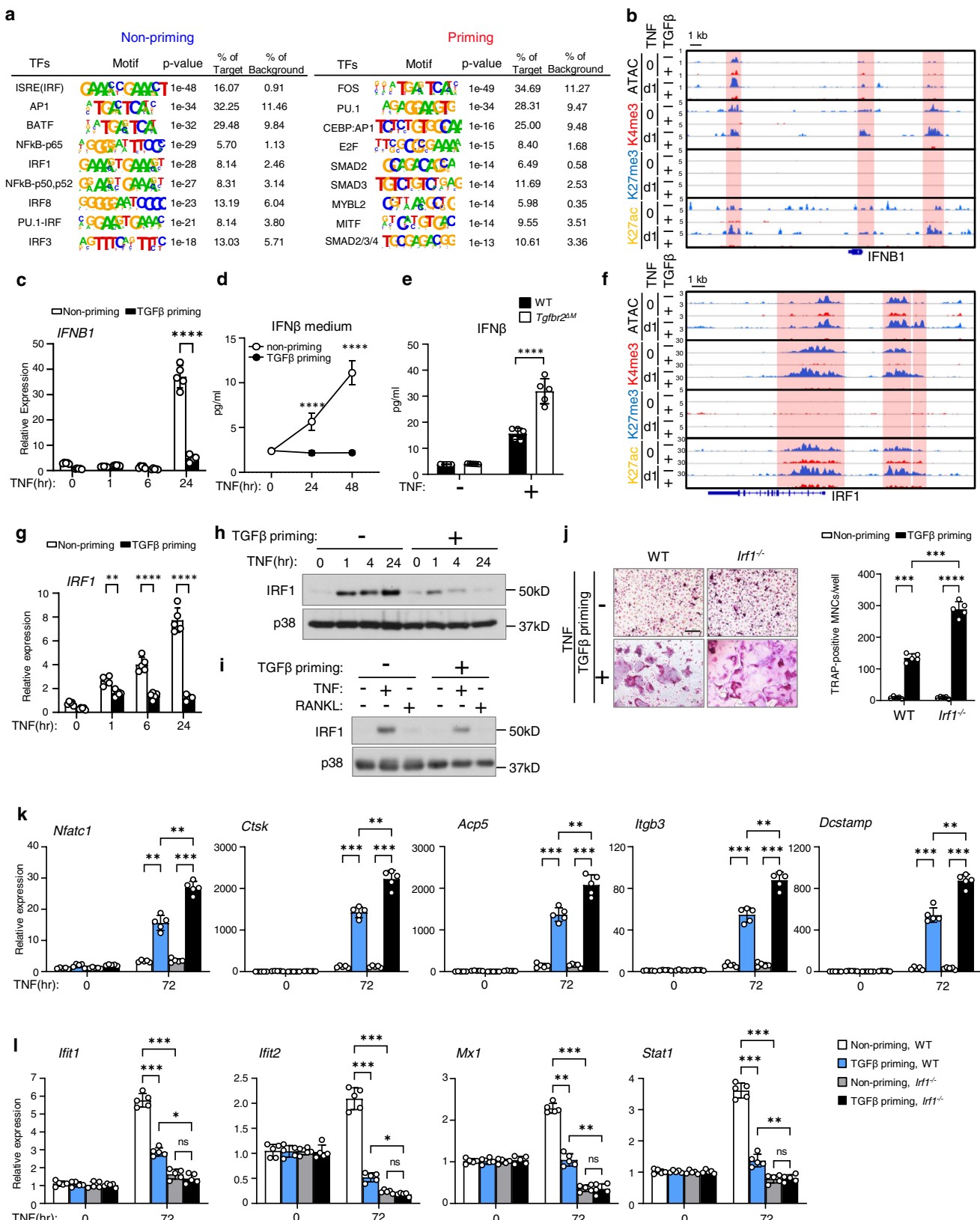

RANKL and TGFβ priming/TNF share some common osteoclastogenic transcription factors, there exist regulators, such as B-Myb, which play a selective role in TGFβ priming/TNF-mediated inflammatory osteoclast differentiation. These results implicate that B-Myb could be a specific therapeutic target for inflammatory osteoclast formation and bone resorption.

**TGFβ expression is highly correlated with osteoclastic gene expression in RA.** Excessive bone erosion by osteoclasts is a critical feature of diseases associated with osteolysis, such as RA. A body of literature has shown that TGFβ expression level is higher in serum and/or synovial fluid in RA patients than in osteoarthritis patients or healthy controls[39–44]. To gain further

**Fig. 5 TNF-induced IRF1-IFNβ-ISG axis is suppressed by TGFβ priming. a** De novo motif-enrichment analysis of ATAC-seq peaks associated with non-priming or TGFβ-priming genes. Random background regions serve as a control. **b** IGV track displaying normalized tag-density profiles for ATAC-seq, H3K4me3, H3K27me3, and H3K27ac Cut&Run-seq signals at *IFNB1* locus. **c** qPCR analysis of mRNA expression of *IFNB1* using human CD14(+)-monocytes treated with or without TGFβ priming for 3 days, followed by TNF stimulation for the indicated times (*n* = 5/group). **d** ELISA analysis of IFNβ levels in the cell culture medium from TNF-induced osteoclastogenesis with or without TGFβ priming (*n* = 5/group). **e** ELISA analysis of IFNβ levels in the serum from the WT and *Tgfbr2ΔM* mice after TNF-induced supracalvarial osteolysis (*n* = 5/group). **f** IGV track displaying normalized tag-density profiles for ATAC-seq, H3K4me3, H3K27me3, and H3K27ac Cut&Run-seq signals at *IRF1* locus. **g, h** qPCR analysis of mRNA expression of *IRF1* (*n* = 5/group) (**g**) and immunoblot analysis of the expression of IRF1 (**h**) using human CD14(+)-monocytes treated with or without TGFβ-priming for 3 days, followed by TNF stimulation for the indicated times. p38 was used as a loading control. **i** Immunoblot analysis of the expression of IRF1 using human CD14(+)-monocytes treated with or without TGFβ-priming for 3 days, followed by TNF or RANKL stimulation for 4 hr. p38 was used as a loading control. **j** Osteoclast differentiation was determined by TRAP staining (left) and the relative area of TRAP-positive-MNCs/well (right) in the cell cultures, in which the bone marrow of WT and *Irf1−/−* mice was primed with or without TGFβ for 4 days, followed by TNF stimulation for two days. TRAP-positive cells: red. (*n* = 5/group). **k–l** qPCR analysis of mRNA expression of the indicated osteoclast genes (**k**) and ISGs (**l**) during osteoclastogenesis using the WT and *Irf1−/−* cells with or without TGFβ-priming followed by TNF stimulation for the indicated times (*n* = 5/group). **c, d, e, g, j, k, l** **p < 0.01; ***p < 0.001; ****p < 0.0001; ns, not statistically significant by two-way ANOVA with Bonferroni's multiple comparisons test. Error bars: **c, d, e, g, j, k, l** Data are mean ± SD. **j** 200 μm. Source data are provided as a Source Data file.

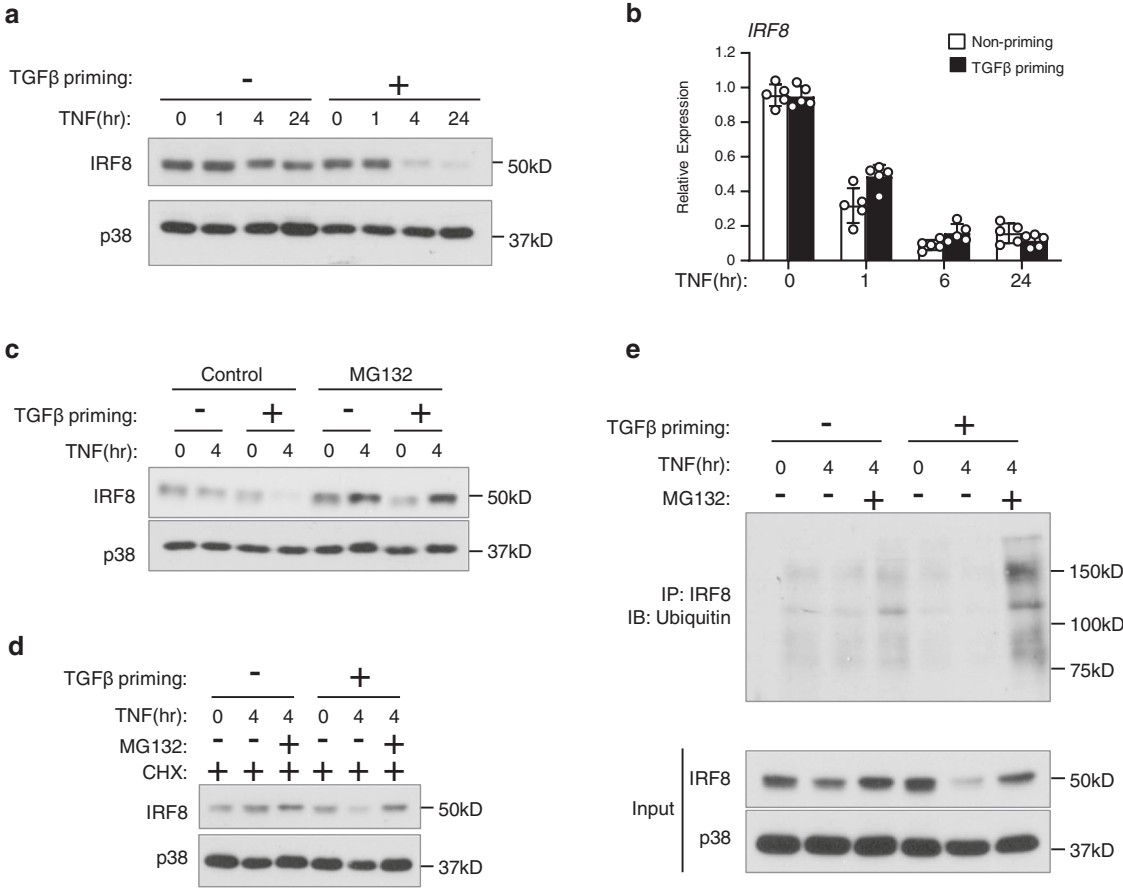

**Fig. 6 TGFβ priming promotes IRF8 ubiquitination and degradation in response to TNF. a, b** Immunoblot analysis (**a**) and qPCR analysis (*n* = 5/group) (**b**) of IRF8 expression using human CD14(+) monocytes treated with or without TGFβ priming for 3 days, followed by TNF stimulation at the indicated times. p38 was used as a loading control (**a**). **c** Immunoblot analysis of the expression of IRF8 using human CD14(+) monocytes treated with or without TGFβ priming for 3 days, followed by TNF stimulation together with DMSO or MG132 (25 μM) for the indicated times. p38 was used as a loading control. **d** Immunoblot analysis of the expression of IRF8 using human CD14(+) monocytes treated with or without TGFβ priming for 3 days, then CHX (50 μM) for 30 min, followed by TNF stimulation together with DMSO or MG132 (25 μM) for the indicated times. p38 was used as a loading control. **e** Ubiquitination of IRF8 in the human CD14(+) monocytes treated with or without TGFβ priming for 3 days, followed by TNF stimulation together with DMSO or MG132 (25 μM) for the indicated times. Cell lysates were immunoprecipitated with anti-IRF8 antibody followed by immunoblotting with anti-Ub antibody. IP, immunoprecipitation; IB, immunoblotting. **b** Data are mean ± SD. Source data are provided as a Source Data file.

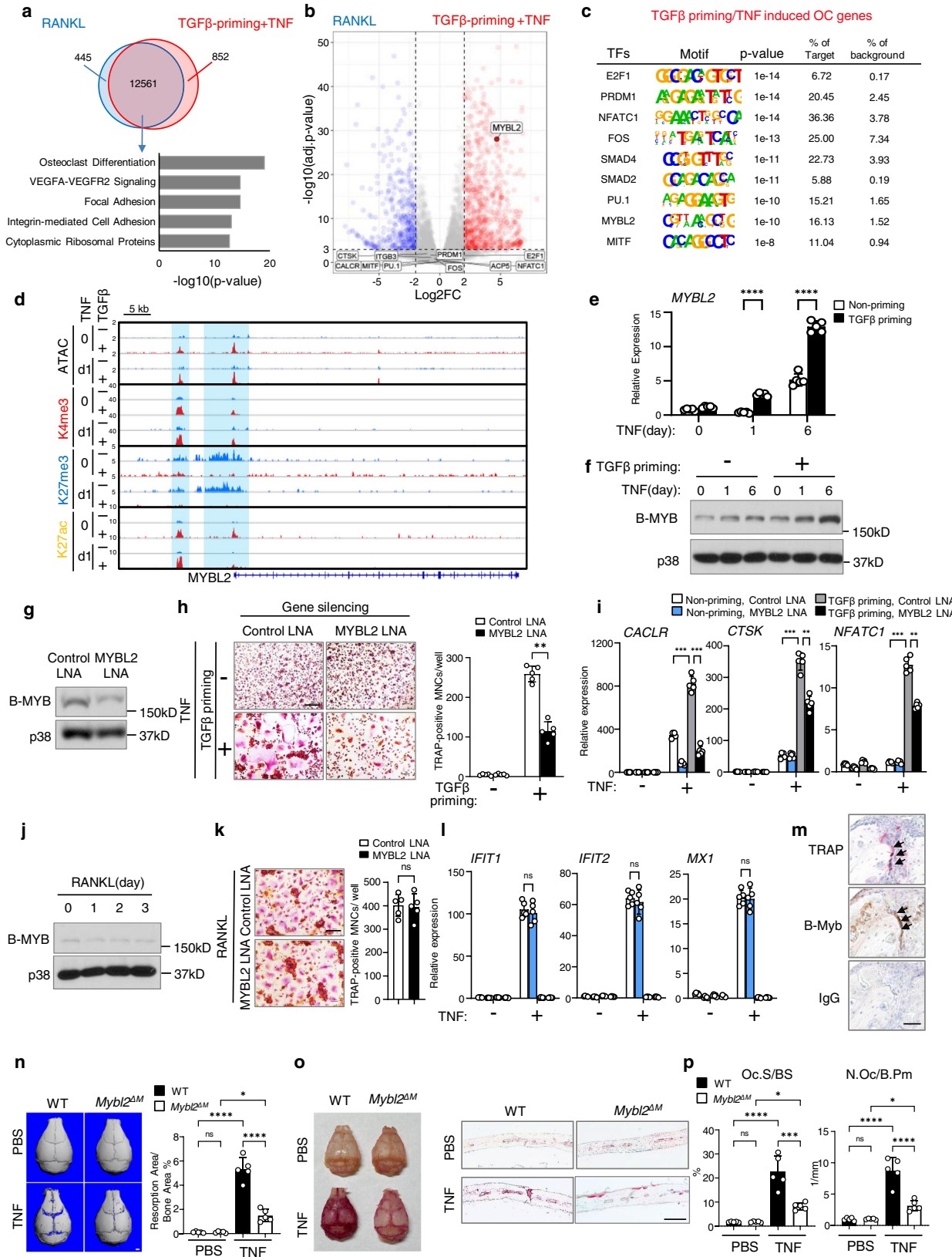

insight into the significance of TGFβ in diseases associated with inflammatory bone erosion, we analyzed the DEG profile by taking advantage of a recently published dataset[60], in which the genome-wide gene expression of PBMCs from cross-sectional cohorts, including 82 SLE patients and 84 RA patients who routinely visited either Brigham and Women's Hospital or Northwell Health during the same time period, was obtained. Although RA and SLE share many symptoms, they are distinct rheumatic diseases. One important distinguishing feature is that RA patients often develop joint erosion with aggressive osteoclast formation/activity, whereas SLE arthropathy is usually non-erosive[61,62]. In addition, type-I IFN signature is common in SLE

**Fig. 7 B-Myb is a previously unrecognized osteoclastogenic regulator specifically involved in TGFβ-priming/TNF-mediated osteoclastogenesis.**
**a** RNA-seq analysis and comparison of mRNA expression induced by RANKL or TGFβ-priming/TNF in human CD14(+)-monocyte cultures. Bottom: pathway analysis of non-DEGs between RANKL and TGFβ-priming/TNF conditions. **b** Volcano plot of the DEGs from **a**. Blue dots: genes more highly expressed in RANKL-induced-condition; red dots: genes more highly expressed in TGFβ-priming/TNF-condition (adjusted $p < 0.001$ and FC > 4). **c** De novo-motif-enrichment analysis of ATAC-seq peaks associated with OC genes in TGFβ-priming/TNF condition. **d** IGV track displaying the indicated seq signals at *MYBL2* locus. **e, f** qPCR analysis of *MYBL2* expression ($n = 5$/group, **e**) and immunoblot analysis of B-Myb (**f**) in human CD14(+)-monocytes treated with/without TGFβ-priming for 3 days, followed by TNF stimulation. **g** Immunoblot analysis of B-Myb in human CD14(+)-monocyte-derived macrophages transfected with LNAs. **h, i, l** TRAP staining and relative area of TRAP-positive-MNCs/well (**h**), and qPCR analysis of the indicated gene expression (**i, l**) in human CD14(+)-monocyte cultures treated with/without TGFβ for 3 days, transfected with the indicated LNAs, and followed by TNF stimulation for 6 days (**h, i**) or 1 day (**l**). ($n = 5$/group). **j** Immunoblot analysis of B-Myb in human CD14(+)-monocyte-derived macrophages stimulated with RANKL. p38 was used as a loading control (**f, g, j**). **k** TRAP staining and relative area of TRAP-positive MNCs/well in human CD14(+)-monocyte-derived macrophages transfected with the indicated LNAs, followed by 3-day-RANKL stimulation. ($n = 5$/group) **m** TRAP (upper) and Immunohistochemical staining of B-Myb (middle, brown) on calvarial slices from TNF-induced osteolysis model in 12-week-old-male mice. Nuclei: blue. Arrows: osteoclasts. $n = 3$. **n–p** μCT images and the quantification of resorption area (**n**), TRAP staining of bone surface (**o**, left) and histological sections (**o**, right), and histomorphometric analysis (**p**) of the calvarial slices from 12-week-old-male mice after PBS or TNF injection to calvarial periosteum daily for 5 days ($n = 5$/group). **e, h, i, l, n, p** *$p < 0.05$; **$p < 0.01$; ***$p < 0.001$; ****$p < 0.0001$; ns, not statistically significant by two-way ANOVA with Bonferroni's multiple comparisons test. **k** ns, by two-sided Student's $t$ test. Error bars: **e, h, i, k, l, n, p** Data are mean ± SD. Scale bars: **h, k** 200 μm; **m** 10 μm; **n** 1.0 mm; **o** 100 μm. Source data are provided as a Source Data file.

patients, and recent literature noted reduced serum TGFβ level in SLE patients[63–65]. Thus, the gene sets from SLE and RA cohorts in this published study appeared to be optimal for us to investigate and compare the correlation between type-I IFN, TGFβ level, and osteoclastic bone erosion in patients. Pathway analysis of the DEGs in RA and SLE revealed distinct associated pathways (Fig. 8a–c). As expected, the highest enriched pathway in SLE is IFN type-I signaling pathway, which is followed by TNF signaling pathway as the second-highest ranked pathway (Fig. 8b). In contrast, the most significantly associated pathway with RA is TGFβ signaling (Fig. 8c). The gene expression heatmaps (Fig. 8d) and GSEA analysis (Fig. 8e) further corroborated the significant enrichment of ISGs (type-I IFN response genes and chemokine genes) with SLE, and enrichment of genes in the TGFβ signaling pathway with RA. Importantly, these analyses revealed highly significantly enriched OC genes in RA PBMCs, but not those from SLE (Fig. 8d, e). The gene expression profiles obtained from SLE and RA PBMC genes almost exactly recapitulated the findings from TNF-induced non-priming gene induction and TGFβ priming gene expression profile, respectively, in our model system. The TGFβ priming condition in our model system likely reflect, at least in part, the role of TGFβ signaling pathway in RA patients. Furthermore, we calculated the average expression level of three typical osteoclastic genes, *NFATC1*, *CTSK* and *ITGB3*, in each patient and used this value to reflect individual osteoclast activity. To indicate the activity of TGFβ signaling pathway in each patient, the average value of the expression of TGFβ and its receptors *TGFBR1* and *2* was calculated. We found that the TGFβ activity was positively correlated with the osteoclast activity in each patient, and impressively both TGFβ activity and osteoclast activity in the RA cohort were much higher than those in the SLE cohort (Fig. 8f). Moreover, consistent with the results in our model system, the mRNA expression levels of *TGFB1*, *FOXM1* and *MYBL2* were significantly higher in RA patients than in SLE patients (Fig. 8g). In contrast, the expression levels of *IFNB1*, *IRF1*, and *IRF8* were reduced in RA compared to SLE patients (Fig. 8g). Taken together, these patient results further support the medical relevance of the correlation between TGFβ signaling and differential expression of ISG and OC genes, and highlight the biological significance of TGFβ signaling in the suppression of inflammation and promotion of osteoclastogenesis in disease settings.

## Discussion

TNF plays important role in immunity and inflammation. A major function of TNF in macrophages is to induce an inflammatory response. In chronic inflammatory diseases associated with bone destruction, such as RA and periodontitis, TNF often stimulates other cell types, such as stromal cells, synovial fibroblasts, and T cells, to aggravate pathologic bone erosion[19]. Interestingly, however, TNF alone is not a potent osteoclastogenic inducer of macrophages. The well-known knowledge is that TNF acts on macrophages to promote their differentiation to osteoclasts mainly through a synergistic action with RANKL[6,10,15–19,66]. Cytokine interactions have important influences on their activities, particularly in various pathological conditions, which form different cytokine networks to affect pathogenesis. Many of the TNF actions occur in combination with other cytokines[51,67]. It is unclear whether cytokines other than RANKL influence the osteoclastogenic ability of TNF. In this study, we discovered a function of TGFβ in the regulation of TNF action on macrophages. TGFβ priming reprograms the macrophage inflammatory response to TNF, switches macrophage cell fate towards osteoclasts, and enables TNF to fully induce osteoclast differentiation independent of RANKL. Except for RANKL, TGFβ is an exclusively identified cytokine that enables TNF to fully induce macrophages to differentiate into osteoclasts to date. Underlying these cellular changes, TGFβ priming has a strong impact on TNF-mediated transcriptome. TNF-induced inflammatory gene expression, including type-I IFN response genes and most chemokine genes, is almost completely turned off, whereas osteoclastic gene expression, including classic and TNF-specific induced osteoclastic genes, is highly turned on by TGFβ priming. TGFβ priming regulates these transcriptomic changes by remodeling chromatin accessibility and histone modifications coordinated with multiple specific transcription factors. Thus, these findings unveil TGFβ as a potent cytokine switch for different TNF actions on macrophages to determine cell fate and function.

This study shows that TGFβ potently influences gene expression profile and macrophage cell fate by switching the TNF role in inflammatory gene induction to osteoclastic differentiation-associated transcriptional program. These transcriptome changes are preceded by TGFβ-induced chromatin remodeling and histone modification. For example, TGFβ attenuates chromatin accessibility and reduces H3K4me3/H3K27ac modification at ISG loci, while opening chromatin access, increasing the levels of H3K4me3/H3K27ac marks and abolishing repressive H3K27me3 marks at OC loci. Notably, these chromatin changes occur before TNF stimulation. Therefore, in our system, prior TGFβ treatment primes chromatin to allow subsequent TNF stimulation to execute osteoclast differentiation programs. In macrophages, most ISG promoters seem to remain in a basal active state, featured

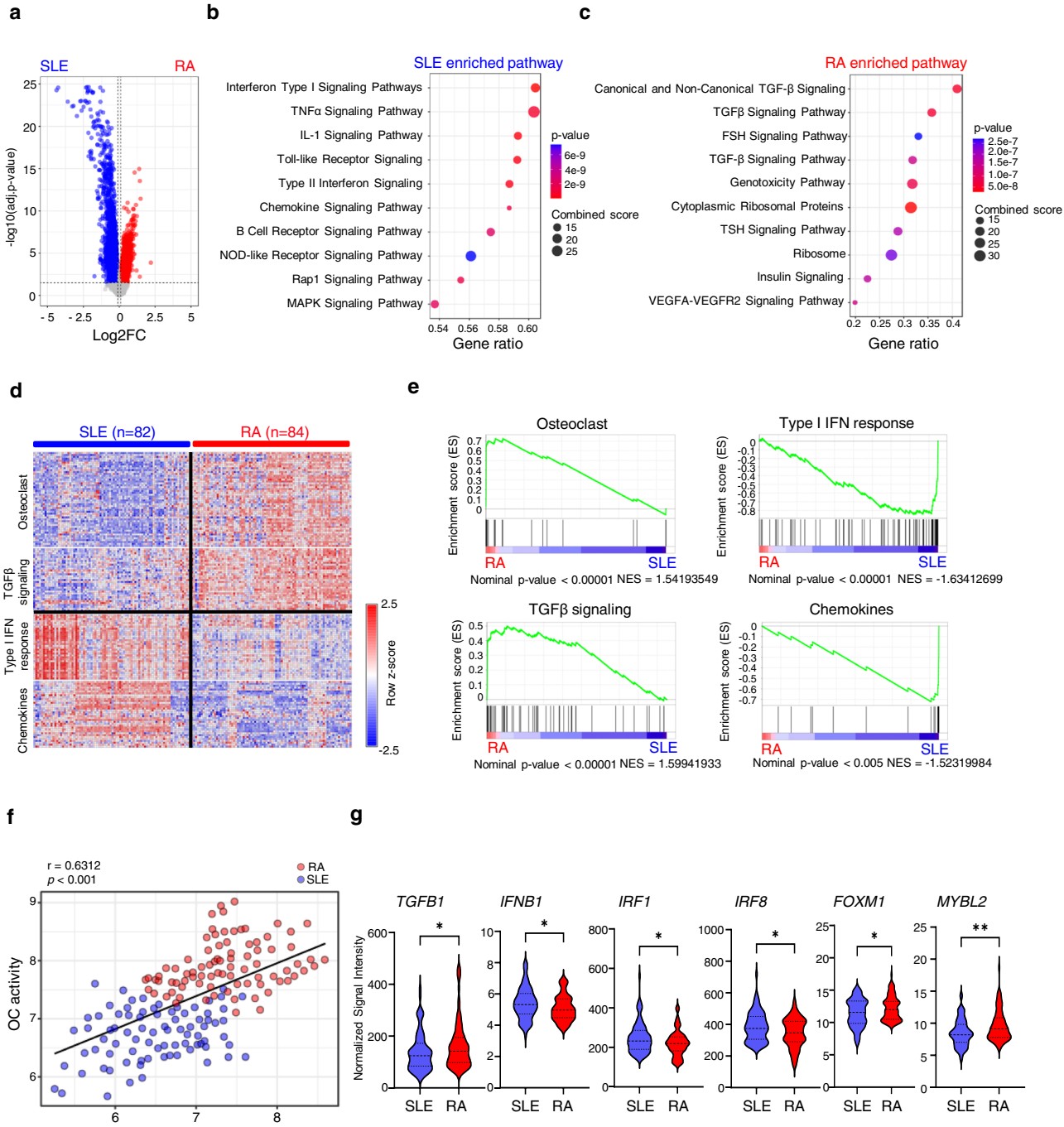

**Fig. 8 Distinct TGFβ level/activity contributes to different osteoclastic activity in RA and SLE patients. a** Volcano plot of microarray analysis of the mRNA expression in the PBMCs isolated from SLE and RA patients. Blue dots: DEGs are more highly expressed in SLE PBMCs. Red dots: DEGs are more highly expressed in RA PBMCs. **b, c** Pathway analysis of the enriched DEGs in SLE (**b**) and RA (**c**). Note: **c** the upper TGFβ signaling Pathway ID is hsa04350 and the lower TGF-β Signaling Pathway ID is WP366. **d** Heatmaps of mRNA expression of the genes involved in osteoclasts, TGFβ signaling, Type-I IFN response, and Chemokine genes in the SLE and RA PBMCs. Row z-scores of Normalized Signal Intensity were shown in the heatmaps. $n = 82$ for SLE patients and $n = 84$ for RA patients. **e** Gene set enrichment analysis of DEG set (osteoclast, TGFβ signaling, type-I IFN response, and chemokine) in SLE and RA PBMCs. **f** Scatter plot showing the significantly positive correlation between the osteoclast activity and TGFβ activity. **g** Normalized Signal Intensity of *TGFB1, IFNB1, IRF1, IRF8, FOXM1,* and *MYBL2* in SLE and RA PBMCs obtained from microarray data. $n = 82$ for SLE patients and $n = 84$ for RA patients. **g** *$p < 0.05$, **$p < 0.01$ by Welch's $t$ test (two-sided). Source data are provided as a Source Data file.

with accessible chromatin and/or the presence of H3K4me3/ H3K27ac mark, which facilitate the fast expression of these genes in response to extracellular stimulations, such as TNF. Most (73%) OC gene loci in macrophages, on the contrary, exhibit a generally repressed state with compacted chromatin and

H3K27me3 mark. Interestingly, the promoters and enhancers of NFATC1 show bivalent domains marked by both H3K4me3 and H3K27me3[68]. This bivalent feature indicates an inactive yet "poised" chromatin state that is resolved to become active or remain repressed along cell differentiation or lineage

specification[68,69]. Indeed, NFATC1 is a master transcription factor for osteoclast lineage differentiation[56]. The poised chromatin state at the NFATC1 locus points to the osteoclastogenic potential of macrophages. TGFβ treatment turns the poised or repressed chromatin states at OC loci into active, meanwhile closing ISG loci in macrophages. Through these actions, TGFβ creates a favorable chromatin environment for TNF to induce macrophages to differentiate into osteoclasts, instead of polarizing to inflammatory macrophages. Following the chromatin remodeling, a number of transcription factors, such as NFATC1, FOS, PRDM1, and B-Myb, are induced and recruited to their target gene loci to join the concert to drive macrophages to differentiate into osteoclasts. On the other hand, the closed promoter and enhancer regions of IRF1 by TGFβ priming results in drastically suppressed transcription and protein expression of transcription factor IRF1 in response to TNF. The diminishment of IRF1–IFNB axis together with the closing of ISG loci eventually turns off ISG expression in macrophages. This whole process provides a paradigm of a signal relay model, in which gene activation and suppression usually occur in multiple sequential steps from extracellular stimulation, signal transduction, regulation at chromatin level coordinated with transcription factors, gene expression, and final cellular phenotype changes. This relay model can also presumably explain why simultaneous TGFβ and TNF treatment is not able to induce osteoclastogenesis, in which chromatin states at OC loci are not primed for transcription by TNF. It is intriguing how TGFβ relays signals to selectively remodel chromatins at different gene loci in monocytes/macrophages. Smad2/3/4 are key signal transducers and transcription factors in the TGFβ signaling pathway. Smads can directly bind to chromatin DNA, but the binding affinity is weak. Pioneer work showed that Smads mediate transcriptional response on the chromatin level by interacting with nucleosome remodelers, chromatin modifiers, and cell/context-dependent cofactors[55]. These mechanisms may apply to the chromatin changes and histone modifications by TGFβ in the present study. Future studies are expected to extensively investigate and elucidate the specific co-regulators by which TGFβ/Smads remodel chromatin in macrophages.

In the past two decades, a great focus has been placed on RANKL-induced osteoclast differentiation, which includes various signaling pathways and a classic group of osteoclastogenic transcription factors, such as NFATc1, c-Fos, and Blimp1. As the physiological inducer of osteoclastogenesis, RANKL plays a key role in normal bone remodeling and bone homeostatic maintenance. Blockade of RANKL signaling by RANKL or RANK antibodies effectively suppresses osteoclast formation and bone erosion. However, long-term blockade of RANKL signaling can affect bone remodeling and suppress new bone formation for healing[19–22]. Thus, there is an unmet medical need to identify alternative strategies to predominantly control inflammatory bone erosion, with minimal side effects on underlying bone remodeling. The mechanisms that selectively regulate inflammatory osteoclastogenesis have remained obscure. In this study, we identified several unique mechanisms specifically involved in TGFβ and TNF co-mediated osteoclastogenesis, without affecting RANKL-induced osteoclastogenesis in physiological conditions. Belonging to the same TNF superfamily, TNF and RANKL share many major signaling pathways. TGFβ switches TNF to induce macrophages to differentiate from osteoclasts. In this process, despite the that TNF induces many of the same classic osteoclastogenic transcription factors and osteoclast marker genes as those induced by RANKL, there exist regulators that are specifically induced by TNF, such as B-Myb, which is a previously unrecognized positive regulator that exclusively promotes TNF-mediated inflammatory, but not RANKL-induced,

osteoclastogenesis. The other unique mechanism is IRF1, which functions as an inhibitor of TNF-mediated osteoclastogenesis. Findings from our group and others[70] show that IRF1 is dispensable for RANKL-induced osteoclast differentiation. This is not surprising after we found that RANKL does not induce IRF1 expression, whereas TNF induces a large amount of IRF1. Both the positive regulator B-Myb and the inhibitor IRF1 are selectively involved in TNF-mediated inflammatory osteoclast differentiation. Furthermore, the expression level of B-Myb is significantly elevated while IRF1 is downregulated in RA PBMCs, supporting their pathogenic roles in inflammatory diseases associated with bone loss. Because they do not significantly influence RANKL-induced osteoclastogenesis, B-Myb and IRF1 are promising therapeutic targets for inflammatory bone resorption. In addition, we found a drastic downregulation of IRF8 by TNF in the TGFβ priming condition. IRF8 is a well-known inhibitor for RANKL-induced osteoclastogenesis. Downregulation of IRF8 is essential for the differentiation process. However, the mechanisms of IRF8 downregulation between RANKL and TGFβ/TNF stimulations are distinct. RANKL inhibits IRF8 expression mainly through transcriptional suppression[58,71,72]. TGFβ/TNF promotes ubiquitination of IRF8 and subsequently its protein degradation. The underlying mechanisms of IRF8 protein degradation may provide unique therapeutic targets that are not included in RANKL signaling. Thus, B-Myb induction, IRF1 decrease, and IRF8 protein degradation are recognized as TNF-specific induced osteoclastogenic mechanisms, which form an inflammatory osteoclastogenic program. Targeting the mechanisms in this specific program has therapeutic implications in the selective treatment of inflammatory bone destruction, minimizing side effects on bone remodeling.

TGFβ regulates a variety of biological processes in physiological conditions, such as morphogenesis, embryonic development, stem cell differentiation, and immune regulation[73]. Thus, a certain serum TGFβ level (usually < 1.5 ng/ml) is maintained in healthy people to achieve the homeostatic function of TGFβ[43,64,65]. In disease settings, serum TGFβ levels are often altered, such as those elevated in RA patients but reduced in SLE patients[39–44,63–65]. The culture medium contains a low level of TGFβ (1–2 ng/ml) similarly to physiological level. We found that TGFβ levels lower than this physiological range in culture medium do not affect TNF function in macrophages (Supplementary Fig. 2), and TGFβ shows a time-dependent (≥2 days) priming effect on TNF activity, indicating that a biological threshold is required for TGFβ priming effects. This also suggests that TGFβ priming on TNF action in macrophages is a slow biological process rather than a rapid cellular response, such as a rapid immune response to toll-like receptor stimulation occurring within minutes to hours, which is presumably because chromatin remodeling and transcriptomic changes require appropriate responses, in terms of both magnitude and duration, to instruct a series of cellular and molecular reactions and determine cell fate. Our results may indicate a possibility of TGFβ priming threshold in disease conditions. For example, although TNF is a key pathogenic cytokine for both RA and SLE, the osteoclast activity and destructive bone erosion present differently in the two groups of patients. Based on our findings, the elevated TGFβ levels in RA and their priming effects on TNF are highly likely part of the reason for the aggressive osteoclastic bone erosion observed in RA but not in SLE. It is unclear, however, what the range of TGFβ threshold levels is for RA patients to initiate bone erosion, and how early the TGFβ levels are elevated and changed during disease progression. In this study, in order to study the effect of TGFβ on TNF action in macrophages and recapitulate this cytokine interaction in vivo, we analyzed the combined data from

in vitro cell culture system, in vivo genetic mouse models, and patients of RA or SLE. Large cohort studies are expected to address these important questions in the future, which may provide insights into the development of biomarkers related to TGFβ signaling pathway for the prognosis of early bone erosion and its severity.

TNF plays an important role in the pathogenesis of both RA and SLE. However, the TNF action on osteoclastogenesis and bone erosion in these diseases are different. TNF promotes osteoclastogenesis and bone destruction in RA joints, but does not induce osteoclastic bone erosion in the joints of SLE patients. TNF activities are often regulated by other cytokines within networks whereby different settings affect and diversify the functions of each cytokine. One critical difference identified from our study between RA and SLE is the TGFβ level/activity. TGFβ level/activity in RA is significantly higher than not only in healthy people but also SLE patients. This elevated TGFβ level/activity in RA is positively correlated with osteoclast activity. Taken together with the strong TGFβ priming effects on TNF-induced osteoclastogenesis in vitro and in vivo, the different bone damage in RA and SLE is attributed, at least partially, to the differences in TGFβ levels or activity. This also reflects the impact of different cytokine interactions on the pathogenesis and consequence of diseases. Genetic evidence from this study shows that a lack of TGFβ signaling strongly prevents inflammatory joint erosion in an inflammatory arthritis model. Recently, the blockade of TGFβ signaling has shown promising therapeutic potential in several clinical trials, such as in cancer treatment[54,74]. For inflammatory diseases associated with bone loss, targeting the osteoclastogenic mechanisms mediated by TGFβ and TNF may provide complementary or alternative strategies to inhibit inflammatory bone destruction and protect bone.

In summary, we found that TGFβ reprograms the macrophage response to TNF and shifts cell fate towards osteoclastogenesis by remodeling chromatin and histone modification in concert with specific transcription factors. These findings identify a previously unrecognized function of TGFβ in the regulation of TNF action on macrophage polarization/differentiation and a RANKL-independent osteoclast differentiation pathway. These mechanisms discovered in the TGFβ and TNF-mediated inflammatory osteoclastogenic program implicate therapeutic strategies to selectively suppress inflammatory bone resorption without significant impact on physiological bone remodeling.

## Methods

**Animal study and analysis of bone phenotype.** We generated mice with myeloid/macrophage-specific deletion of *Tgfbr2* by crossing the *Tgfbr2^flox/flox^* mice (The Jackson Laboratory, Stock No: 012603) with the mice with a lysozyme M promoter-driven Cre transgene on the C57BL/6 background (known as LysMcre; The Jackson Laboratory, Stock No: 004781). Gender- and age-matched *Tgfbr2^flox/flox^;LysMcre(+)* mice (referred to as *Tgfbr2^ΔM^*) and their littermates with *Tgfbr2^+/+^;LysMcre(+)* genotype as WT controls (hereafter referred to as WT) were used for experiments. *Irf1* knockout mice (referred to as *Irf1^-/-^*) were obtained from the Jackson Laboratory (Stock No: 002762). The bone marrow isolated from gender- and age-matched *Irf1^-/-^* and WT (*Irf1^+/+^*) were used in the experiments. We generated mice with myeloid/macrophage-specific deletion of *Mybl2* by crossing the *Mybl2^flox/flox^* mice[59] with the LysMcre mice. Gender- and age-matched *Mybl2^flox/flox^;LysMcre(+)* mice (referred to as *Mybl2^ΔM^*) and their littermates with *Mybl2^+/+^;LysMcre(+)* genotype as WT controls (hereafter referred to as WT) were used for experiments.

For inflammatory osteolysis experiments, we used the established TNF-induced supracalvarial osteolysis mouse model with minor modifications[47,48]. TNFα was administrated daily at the dose of 75 μg/kg to the calvarial periosteum of age- and gender-matched mice for 5 consecutive days or 14 consecutive days before the mice were sacrificed. The calvarial, femoral, and L5 vertebral bones were fixed and subjected to micro-computed tomography (μCT) analysis, sectioning, TRAP staining, and histological analysis.

For Inflammatory arthritis experiments, we used K/BxN Serum Transfer-Induced Arthritis model[49]. K/BxN serum pools were prepared, and arthritis was induced by intraperitoneal injection of 100 μl of K/BxN serum to the male mice on days 0 and 2. The development of arthritis was monitored by measuring the thickness of wrist and ankle joints with digital slide caliper (Bel-Art Products). For each animal, the joint thickness was calculated as the sum of the measurements of both wrists and ankles. Joint thickness was represented as the average for each group. Mice were sacrificed on day 14 and serum and paws were collected. Hind paws were subjected to μCT analysis, sectioning, TRAP staining and histological analysis. μCT analysis of femoral trabecular bones, cortical midshaft, and L5 vertebral trabecular bones was conducted to evaluate the bone volume and 3D bone architecture using a Scanco μCT-35 scanner (SCANCO Medical) according to the manufacturer's instructions and the American Society of Bone and Mineral Research[75] guidelines. All mice were housed in a 12-h light cycle at room temperature and had dry laboratory food and water ad libitum. All animal procedures were approved by the Hospital for Special Surgery Institutional Animal Care and Use Committee (IACUC), and Weill Cornell Medical College IACUC.

**Reagents.** Murine or human M-CSF, murine or human TNFα, human TGFβ1, and soluble human RANKL were purchased from PeproTech. Murine TGFβ1 was purchased from R&D systems. MG132, and Cycloheximide (CHX) were purchased from Millipore.

**Cell culture.** For human cell cultures, de-identified blood buffy coats (blood leukocyte preparations) were purchased from the New York Blood Center using a protocol approved by the Hospital for Special Surgery (HSS) Institutional Review Board. The blood buffy coats were anonymous without any identifiable private information. As per Human Subjects Research in PHS SF424 (R&R) Application Guide, studies using purchased de-identified blood samples do not constitute human subject research; informed consent was not obtained at HSS. PBMCs from the buffy coats were isolated by density gradient centrifugation using Ficoll (Invitrogen Life Technologies, Carlsbad, CA), and CD14(+) cells were purified from fresh PBMCs using anti-CD14 magnetic beads (Miltenyi Biotec, Auburn, CA) according to the manufacturer's instructions. Human CD14(+) monocytes were cultured at a density of $15.6 \times 10^4/\text{cm}^2$ in α-MEM medium (Thermo Fisher Scientific) containing 10% FBS (Atlanta Biologicals), glutamine (2.4 mM, Thermo Fisher Scientific), and Penicillin–Streptomycin (Thermo Fisher Scientific) in the presence of M-CSF (20 ng/ml; PeproTech, Rocky Hill, NJ) with or without human TGFβ1 (10 ng/ml; PeproTech, Rocky Hill, NJ) for 3 days to induce macrophages with or without TGFβ priming, respectively. The cells were then washed with neat α-MEM medium to remove TGFβ, and further cultured with TNFα (40 ng/ml) and M-CSF (20 ng/ml) in the α-MEM medium for different times indicated in figure legends.

For mouse cell cultures, mouse bone marrow cells were harvested from the tibiae and femora of the age- and gender-matched mutant and control mice and cultured for 3 days in α-MEM medium containing 10% fetal bovine serum (FBS), glutamine (2.4 mM, Thermo Fisher Scientific), Penicillin–Streptomycin (Thermo Fisher Scientific) and CMG14–12 supernatant (the condition medium, which contained the equivalent of 20 ng/ml of rM-CSF and was used as a source of M-CSF) with or without mouse TGFβ1 (1 ng/ml; R&D systems, Minneapolis, MN). The attached BMMs were scraped, seeded at a density of $4.5 \times 10^4/\text{cm}^2$, and cultured in α-MEM medium with 10% FBS, 1% glutamine and the condition medium for overnight. Except where stated, the cells were then treated without or with optimized concentrations of TNFα (40 ng/ml) in the presence of the condition medium for times indicated in the figure legends. Culture media were exchanged every 3 days.

TRAP staining was performed with an acid phosphatase leukocyte diagnostic kit (Sigma-Aldrich) in accordance with the manufacturer's instructions. TRAP-positive cells show red staining.

**Mineral resorption pit assay.** The mineral resorption activity of osteoclasts was examined using 96-well Corning Osteo Assay Surface Plates (Sigma-Aldrich). Human CD14(+) monocytes were seeded at a density of $15.6 \times 10^4/\text{cm}^2$ in Osteo Assay Surface Plate and cultured in the presence of M-CSF with or without TGFβ for 3 days, then TGFβ was washed away, and the cells were further cultured with TNF (40 ng/ml) for 10 days. Cells were then removed twice with 10% bleach solution for 5 min at room temperature (RT), followed by washing with distilled water. The minerals were stained with Von Kossa to visualize the formation of resorptive pits. The resorptive area was analyzed using ImageJ (National Institutes of Health, Bethesda, MD, USA).

**Phagocytosis assay.** Phagocytosis assay was performed by using Zymosan A (*S. cerevisiae*) BioParticles™, Alexa Fluor™ 488 conjugate (Thermo Fisher Scientific Z23373). Zymosan was incubated in cell cultures at a final concentration of 2.5 μg/ml for 20 min at 37 °C. The cells were fixed with 4% PFA for 30 min at RT, permeabilized with 0.5% TritonX-100/PBS for 5 min at RT, and then stained with DAPI (1:1000) and Alexa Fluor® 555 Phalloidin (1:20, #8953, Cell Signaling Technology). Images were obtained by ZEISS Axio Observer 7 (Zeiss) and analyzed by ImageJ.

**In vitro gene silencing.** Antisense inhibition using locked nucleic acid (LNA) technology from Qiagen was applied to silence gene expression in vitro. LNA

oligonucleotides specifically targeting *MYBL2* and non-targeting control LNAs were from Qiagen and were transfected into human CD14(+) monocytes derived macrophages at concentrations of 40 nM using TransIT-TKO transfection reagent (Mirus) in accordance with the manufacturer's instructions.

**Reverse transcription and real-time PCR**. DNA-free RNA was extracted from cells using the RNeasy Mini Kit (Qiagen) with DNase treatment. Reverse transcription was performed using 1 µg of total RNA with random hexamers and MMLV-Reverse Transcriptase (Thermo Fisher Scientific) according to the manufacturer's instructions. Real-time PCR was performed in triplicate with the QuantStudio 5 Real-time PCR system and Fast SYBR® Green Master Mix (Thermo Fisher Scientific) with 500 nM primers[76]. mRNA amounts were normalized relative to glyceraldehyde-3-phosphate dehydrogenase (*GAPDH*) mRNA. The primers for real-time PCR were as follows: *Nfatc1*: 5′-CCCGTCACATTCTGGTCCAT-3′ and 5′-CAAGTAACCGTGTAGCTCCACAA-3′; *Prdm1*: 5′-TTCTTGTGTGGTATTG TCGGGACTT-3′ and 5′-TTGGGGACACTCTTTGGGTAGAGTT-3′; *Acp5*: 5′-A CGGCTACTTGCGGTTTC-3′ and 5′-TCCTTGGGAGGCTGGGTC-3′; *Ctsk*: 5′-AA GATATTGGTGGCTTTGG-3′ and 5′-ATCGCTGCGTCCCTCT-3′; *Itgb3*: 5′-CCG GGGGACTTAATGAGACCACTT-3′ and 5′-ACGCCCCAAATCCCACCCATA CA-3′; *Dcstamp*: 5′-TTTGCCGCTGTGGACTATCTGC-3′ and 5′-AGACGTGGT TTAGGAATGCAGCTC-3′; *Fos*: 5′-AGACCAGAGCGCCCCATCCTTACG-3′ and 5′-GCTCTGCGCTCTGCCTCCTGACA-3′; *Mx1*: 5′-GGCAGACACCACAT ACAACC-3′ and 5′-CCTCAGGCTAGATGGCAAG-3′; *Ifit1*: 5′-CTCCACTTTCA GAGCCTTCG-3′ and 5′-TGCTGAGATGGACTGTGAGG-3′; *Ifit2*: 5′-AAATGTC ATGGGTACTGGAGTT-3′ and 5′-ATGGCAATTATCAAGTTTGTGG-3′; *Gapdh*: 5′-ATCAAGAAGGTGGTGAAGCA-3′ and 5′-AGACAACCTGGTCCTCAGTGT-3′; *NFATC1*: 5′-AAAGACGCAGAAACGACG-3′ and 5′-TCTCACTAACGGGAC ATCAC-3′; *CALCR*: 5′-CTGAAGCTTGAGCGCCTGAGTC-3′ and 5′-TGGGGTT GGGTGATTTAGAAGAAG-3′; *ITGB3*: 5′-GGAAGAACGCGCCAGAGCAAAA TG-3′ and 5′-CCCCAAATCCCTCCCCACAAATAC-3′; *FOS*: 5′-GCAAGGTGGA ACAGGAGACA-3′ and 5′-CAGATCAAGGGAAGCCACAG-3′; *MYBL2*: 5′-TGG CTTTGCCTATGTGGA-3′ and 5′-CCTGGTCTTAAAGAGGGACTT-3′; *CTSK*: 5′-CTCTTCCATTTCTTCCACGAT-3′ and 5′-ACACCAACTCCCTTCCAAAG-3′; *IL1B*: 5′-TTCGACACATGGGATAACGAGG-3′ and 5′-TTTTTGCTGTGAG TCCCGGAG-3′; *IL6*: 5′-TAATGGGCATTCCTTCTTCT-3′ and 5′-TGTCCTAA CGCTCATACTTTT-3′; *IRF1*: 5′-CAAATCCCGGGGCTCATCTG-3′ and 5′-CT GGCTCCTTTTCCCCTGCTTTCT-3′; *IRF8*: 5′-TGCGCTCCAAACTCATTCTC-3′ and 5′-TGGAAACATCCGGAAGACCTG-3′; *IFIT1*: 5′-TTCGGAGAAAGGCA TTAGA-3′ and 5′-TCCAGGGCTTCATTCATAT-3′; *IFIT2*: 5′-CGCAGTGCAGC CAAGTTTTATC-3′ and 5′-GCAGGTAGGCATTGTTTGGTAT-3′; *MX1*: 5′-AG CCACTGGACTGACGACTT-3′ and 5′-ACCACGGCTAACGGATAAG-3′; *STAT1*: 5′-CAGCTTGACTCAAAATTCCTGGA-3′ and 5′-TGAAGATTACGCT TGCTTTTCCT-3′; *IFNB1*: 5′-AGAAGCTCCTGTGGCAATTG-3′ and 5′- ACTGC TGCAGCTGCTTAATC-3′; *CCL5*: 5′-GAGGCTTCCCCTCACTATCC-3′ and 5′- CTCAAGTGATCCACCCACCT-3′; *CXCL9*: 5′-CTGTTCCTGCATCAGCACCA AC-3′ and 5′-TGAACTCCATTCTTCAGTGTAGCA-3′; *CXCL10*: 5′-ATTTGCT GCCTTATCTTTCTG-3′ and 5′-TCTCACCCTTCTTTTTCATTGTAG-3′; *MYBL2*: 5′-ACAGGTGGCTGAGAGTTTTG-3′ and 5′-TTCAGGTGCTTGGCAA TCAG-3′; *GAPDH*: 5′-ATCAAGAAGGTGGTGAAGCA-3′ and 5′-GTCGCTGTT GAAGTCAGAGGA-3′.

**Immunoblot analysis**. Total cellular extracts were obtained using lysis buffer containing 150 mM Tris-HCl (pH 6.8), 6% SDS, 30% glycerol, and 0.03% Bromophenol Blue; 10% 2-ME was added immediately before harvesting cells. Cell lysates were fractionated on 7.5% SDS-PAGE, transferred to Immobilon-P membranes (Millipore), and incubated with specific antibodies. Western Lightning plus-ECL (PerkinElmer) was used for detection. NFATc1 antibody (556602, 1:1000) was from BD Biosciences; Blimp1 (sc-47732, 1:1000), c-Fos (sc-52, 1:1000), IRF8 (sc-6058, 1:1000), and p38α (sc-535, 1:3000), B-Myb (sc-390198, 1:1000) antibodies were from Santa Cruz Biotechnology; IRF1 (8478, 1:1000) antibody was obtained from Cell Signaling Technology.

**ELISA**. Mouse serum IFNβ and culture medium IFNβ were measured by using VeriKine-HS Mouse Interferon Beta Serum ELISA Kit (PBL Assay Science) according to the manufacturer's instruction.

**Immunoprecipitation**. Cells (10,000,000 cells/condition) were washed with ice-cold PBS and lysed for 15 min on ice with lysis buffer (50 mM Tris-HCl, pH 8.0, 280 mM NaCl, 0.5% NP-40, 0.2 mM EDTA, and 10% glycerol) containing protease inhibitors (1× Complete™ Protease Inhibitor Cocktail (Roche) and 10 mM phenylmethylsulfonyl fluoride (PMSF)). Cell lysates were centrifuged at 14,000 rpm for 15 min at 4 °C. 5% supernatant was used as input. The leftover supernatant was incubated with anti-IRF8 antibody (6 µg, sc-6058, Santa Cruz Biotechnology) overnight at 4 °C and then incubated with Dynabeads™ Protein G (50 µl per sample, Thermo Fisher Scientific) for 4 hr at 4 °C for precipitation. The beads were washed five times with the lysis buffer and subjected to immunoblotting with anti-Ubiquitin antibody (sc-8017, 1:1000, Santa Cruz Biotechnology).

**RNA-seq and analysis**. Total RNA was extracted from cultured primary human macrophages using RNeasy Mini Kit (QIAGEN) following the manufacturer's instructions. NEBNext Ultra II RNA Library Prep Kit for Illumina (NEB) was used to purify poly-A + transcripts and generate libraries with multiplexed barcode adaptors following the manufacturer's instructions. All samples passed quality control analysis using a Bioanalyzer 2100 (Agilent). High-throughput sequencing (50 bp, single-end) was performed using the Illumina Hiseq 4000 in the Weill Cornell Medicine Genomics Resources Core Facility with a sequencing depth between 30 to 50 million reads per sample. RNA-seq reads were aligned to the human genome (GRCh38) using HISAT2 with default parameters[77]. Reads were counted by HTseq-count[78] and edgeR[79] was used to estimate the transcript abundances as counts per million (CPM) values and calculate adjusted $p$ value (adj.$p$ value) and log2 fold-change (Log2FC). Genes with low expression levels (<1 CPM) in all conditions were filtered from downstream analyses. Benjamini-Hochberg false discovery rate (FDR) procedure was used to correct for multiple testing. Genes with adjusted $p$ value <0.05 and fold-change of at least 1.5 were identified as DEG between conditions from the edgeR analysis of three RNA-seq biological replicates from different donors. Fastq files of the RNA-seq data using cultured primary human macrophages stimulated with RANKL for two days to induce osteoclastogenesis were extracted from GSE171542[80]. The reads were aligned to the human genome (GRCh38) using HISAT2 with default parameters. HTseq-count was subsequently used to count reads and then edgeR was used to estimate the transcript abundances as CPM (counts per million) values and calculate adj.$p$ value and Log2FC. Genes with low expression levels (<1 CPM) in all conditions were filtered from downstream analyses. Genes with adjusted $p$ value <0.001 and log$_2$(fold-change) of at least 2 were identified as DEGs between RANKL and TGFβ priming/TNF conditions in Fig. 7a, b. Three biological replicates of RNA-seq were used for RANKL-induced mRNA expression in human osteoclast differentiation cultures using CD14(+)-monocytes and the mRNA induction in human CD14(+)-monocytes treated with TGFβ-priming/TNF stimulation in Fig. 7a. The non-overlapped genes in the Venn-diagram show the DEGs in different conditions, and the overlapped genes are non-DEGs in Fig. 7a. The volcano plot was generated by ggplot2 package in R. Heatmaps were generated by the pheatmap package in R. Integrated pathway analysis was performed using KEGG and Wikipathways in IMPaLA (Integrated Molecular Pathway Level Analysis)[81]. Gene Set Enrichment Analysis (GSEA program, Broad Institute) input with the DEGs was performed according to the program's instructions[82]. $p$ and FDR values were calculated following the program's instructions.

**ATAC-seq and analysis**. ATAC-seq was performed according to the Omni-ATAC protocol[83]. In all, 50,000 cells were collected and washed with cold ATAC-Resuspension Buffer (RSB) containing 0.1% NP-40, 0.1% Tween-20, and 0.01% Digitonin. Cells were lysed with cold ATAC-RSB containing 0.1% Tween-20. Pelleted nuclei were incubated with transposition mix (25 µl 2× TD buffer (Illumina), 2.5 µl transposase (Illumina), 16.5 µl PBS, 0.5 µl 1% digitonin, 0.5 µl 10% Tween-20, 5 µl nuclease-free water) for 30 min at 37 °C in a thermomixer at 1000 rpm. Transposed DNA was purified using DNA Clean & Concentrator (Zymo Reserach). We amplified library fragments using previously published barcoded primers[84], with the following PCR conditions: 72 °C for 5 min; 98 °C for 30 s; and thermocycling at 98 °C for 10 s, 63 °C for 30 s, and 72 °C for 1 min for a total of 10–13 cycles. The libraries were purified using DNA Clean & Concentrator. Library quality and quantification were assessed with an Agilent Bioanalyzer at the Weill Cornell Medicine Genomics Resources Core Facility. Barcoded sample libraries were pooled for a final concentration of 4 nM. Sequencing was performed on Illumina Hiseq4000 (50 bp, single-end) at the Weill Cornell Genomics Resources Core Facility. Sequenced reads were aligned to reference the human genome (GRCh38) using Bowtie2 with the default parameter. The read depth was 80 to 90 million reads for each sample. The total number of mapped reads in each sample was normalized to one million mapped reads. Peak calling was performed using MACS2[85] with a $q$ value cutoff of 0.01. Differential accessibility analysis of peaks was performed with Diffbind[86]. Differentially accessible peaks were defined as false discovery rate (FDR) < 0.01 and fold-change of at least 2. The peaks were assigned to each gene locus, including 20 kb upstream of the transcription start site, gene body and 5 kb downstream of transcription termination site. Volcano plot was generated by ggplot2 package in R. For integrative analysis of RNA-seq and ATAC-seq, DEGs from RNA-seq or ATAC-peaks-associated genes were combined from all time points in priming or non-priming condition. The overlapped genes between DEGs and ATAC-peaks-associated genes in priming or non-priming condition were used for downstream analyses in Fig. 4. For visualizing the ATAC-seq data, bigwig files were created from bam files with deeptools[87], normalized using the Counts Per Million mapped reads (CPM) method, and then the peaks were visualized in Integrative Genomics Viewer (IGV)[79]. Tag density was generated by using *annotatePeaks.pl* in HOMER package[88]. Peak density heatmaps and peak profiles were generated by using deeptools[87]. Boxplots were generated by Prism 8 (GraphPad).

**Cut & Run-seq and analysis**. Cut & Run-seq was performed according to a published protocol[89]. In all, 500,000 cells were used per condition. Cells were washed by wash buffer (20 mM HEPES, pH 7.5, 150 mM NaCl, 0.5 mM Spermidine, 1× Complete™ Protease Inhibitor Cocktail (Roche), then mixed with

Concanavalin A beads and permeabilized with Cell Permeabilization Buffer (wash buffer containing 0.01% Digitonin). The cells were then incubated with the primary antibodies (H3K4me3: 07-473, Millipore, 1:100; H3K27me3: 07-449, Millipore, 1:100; H3K27ac: ab4729, Abcam, 1:100) for 16 hr at 4 °C, followed by incubation with protein A-MNase (Cell Signaling Technology) for 1 hr at 4 °C. MNase was activated with CaCl$_2$ for 2 hr at 4 °C. After adding stop buffer (340 mM NaCl, 20 mM EDTA, 4 mM EGTA, 50 μg/mL RNase A, 50 μg/mL Glycogen), samples were incubated for 30 min at 37 °C to release chromatin from cells. Fragmented chromatins were collected and purified with DNA Clean & Concentrator (Zymo Research). Library preparation was performed using NEBNext® Ultra™ II DNA Library Prep Kit according to the manufacturer's instructions. Sequencing was performed on the Illumina NextSeq 500 (50 bp, paired-end). The read depth was 8 to 10 million reads for each condition. Then, sequenced reads were aligned to reference the human genome (GRCh38) using Bowtie2 with the parameter,–*end-to-end–very-sensitive–no-mixed–dovetail–no-discordant–phred33 -I 10 -X 700*. Peak calling was performed using MACS2[85] with a *q* value cutoff of 0.01. Tag density was generated by using *annotatePeaks.pl* in HOMER package[88]. Peak density heatmaps and peak profiles were generated by using deeptools[87]. For visualizing the peaks of histone modifications, bigwig files were created from bam files with deeptools, normalized using the CPM method, and then the peaks were visualized in IGV[79]. Boxplots were generated by Prism 8 (GraphPad).

**Motif-enrichment analysis**. For de novo motif analysis, finding enriched motifs of transcription factors was performed with *findMotifsGenome.pl* in HOMER package, on ±300 bp centered on the ATAC-seq peak. Peak sequences were compared to random genomic fragments of the same size and normalized G + C content to identify motifs enriched in the targeted sequences.

**FAIRE (formaldehyde-assisted isolation of regulatory element)-qPCR**. 1,000,000 cells were used per condition. Cells were treated with 1% formaldehyde for 5 min to crosslink chromatin. Crosslink was quenched by the addition of 0.125 M glycine for 5 min. The cells were washed with cold phosphate-buffered saline and collected. Fixed cells were lysed using buffer LB1 (50 mM HEPES-KOH, pH 7.5, 140 mM NaCl, 1 mM EDTA, 10% glycerol, 0.5% NP-40, 0.25% Triton X-100, and 1× Complete™ Protease Inhibitor Cocktail) for 10 min. Pelleted nuclei were resuspended in buffer LB2 (10 mM Tris-HCl, pH 8.0, 200 mM NaCl, 1 mM EDTA, 0.5 mM EGTA, and 1× Complete™ Protease Inhibitor Cocktail) and incubated for 10 min on ice. The nuclei were pelleted and lysed in buffer LB3 (10 mM Tris-HCl, pH 8.0, 100 mM NaCl, 1 mM EDTA, 0.5 mM EGTA, 0.1% Na-deoxycholate, 0.5% N-lauroylsarcosine, and 1v Complete™ Protease Inhibitor Cocktail). A total of 10% of nuclear lysates were saved as input. Chromatin was sheared using a Bioruptor Pico device (Diagenode). Free chromosomal DNAs were extracted from nuclear lysates by phenol-chloroform and then de-crosslinked for qPCR analysis using specific primers[90]. Chromatin accessibility is displayed relative to total input. Primers are listed in supplementary table 1.

**Immunohistochemistry**. Calvarial bones from 12-week-old-male WT mice administrated with TNF (75ug/kg) via calvarial periosteum daily for 5 days were collected and immediately fixed in 4% paraformaldehyde solution overnight. Bones were decalcified with 0.5 M EDTA (pH 7.5) at 4 °C for 2 days. All samples were embedded in paraffin and sliced into 5-um-thick sections. Sections were deparaffinized and hydrated. Endogenous peroxidase activity of the tissues was quenched by incubating the sections with 3% H$_2$O$_2$ in MetOH for 20 min. After washing sections with PBS, sections were treated with 0.1% trypsin for 30 min at 37 °C. Then, sections were blocked with MOM blocking reagents (Vector Laboratories). Subsequently, sections were incubated with anti-B-Myb antibody (sc-390198, 1:100) or normal mouse IgG (sc-2025, 1:100) at 4 °C overnight followed with biotinylated anti-mouse IgG antibody (Vector Laboratories, BA-9200-1.5, 1:200) at room temperature for 1 hr. The sections were then incubated with VECTASTAIN ABC Reagent (Vector Laboratories) at room temperature for 30 min, and subsequently incubated with diaminobenzidine (DAB, MilliporeSigma). Sections were then counter-stained with hematoxylin for nuclei staining and were mounted with cover slip using Clear-Mount (Electron Microscopy Sciences). IgG was used as a negative control for B-Myb staining. Nuclei stained with hematoxylin show blue.

**Analysis of Gene Expression in PBMCs from RA and SLE patients**. Microarray raw data were extracted from GSE110169[60]. We analyzed the microarray data using the affy and limma package in R[91,92]. Genes with adjusted *p* value <0.05 and FC > 1.1 were identified as DEGs between RA and SLE. Integrated pathway analysis was performed using KEGG and Wikipathways in IMPaLA[81] with gene ratio and *p* value. Combined score was calculated by Enrichr. The dot plots of enriched pathways and volcano plot were generated by ggplot2 package in R. Heatmaps were generated by the pheatmap package in R. Gene Set Enrichment Analysis (GSEA program, Broad Institute) input with the DEGs was performed according to the program's instructions. *p* and FDR values were calculated following the program's instructions.

**Statistical analysis**. Statistical analysis was performed using Graphpad Prism® software. Two-tailed Student's *t* test or Welch's *t* test was applied when there were only two groups of samples. In the case of more than two groups of samples, one-way analysis of variance (ANOVA) was used with one condition, and two-way ANOVA was used with more than two conditions. ANOVA analysis was followed by post hoc Bonferroni's correction for multiple comparisons. *p* < 0.05 was taken as statistically significant. Data are presented as the mean ± SD or ± SEM as indicated in the figure legends.

## Data availability

The data sets that support the findings of this study and were generated by the authors as part of this study have been deposited in the Gene Expression Omnibus database with the accession code: GSE171843. Source data are provided with this paper.

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

## Acknowledgements

We thank Weill Cornell Medicine Genomics Core Facilities for next-generation sequencing, Dr. Franck Barrat for sharing the de-identified human CD14(+) monocytes isolated from the buffy coats purchased from New York Blood Center, Drs. Theresa Lu, Marie Dominique Ah Kioon, Vidyanath Chaudhary, Ruoxi Yuan, Chao Yang, Upneet Sokhi, Mahesh Bachu, Caroline Brauner, and Bikash Mishra for technical assistance, and Courtney Ng for critical review of the manuscript. We are grateful to the lab members from Dr. Baohong Zhao's laboratory for their helpful discussions and assistance. M.B.G. holds a Career Award for Medical Scientists from the Burroughs Welcome Foundation, and a Pershing Square Sohn Prize for Young Investigators in Cancer Research. This work was supported by grants from the National Institutes of Health (AR075585 to MBG, AR068970, AR071463, and AR078212 to B.Z.) and by support for the Rosensweig Genomics Center at the Hospital for Special Surgery from The Tow Foundation. The content of this manuscript is solely the responsibility of the authors and does not necessarily represent the official views of the NIH.

## Author contributions

Y.X. and K.I. designed and performed the experiments, analyzed and curated data, prepared figures, and contributed to manuscript preparation. K.I. performed bioinformatics analysis and curated data. Y.D. isolated human PBMCs and assisted with experiments. E.P.R. and S.J.B. provided Mybl2 flox mice and contributed to manuscript preparation. M.B.G. provided instruction on experimental designs, discussed data, and contributed to manuscript preparation. B.Z. conceived, designed, supervised the project, and wrote the manuscript. All authors reviewed, provided input on the manuscript, and approved submission.

## Competing interests

The authors declare no competing interests.
