## [Peer Review File · Nature Communications]

Reviewers' Comments:

Reviewer #1:

Remarks to the Author:

Xia and colleagues have studied the role of TGFb/TNF-driven inflammatory osteoclastogenic program in excessive bone resorption condition. This interesting paper shows a novel function of TGFb in the regulation of TNF action on osteoclast differentiation via RANKL-independent pathway. The data is solid, but I would like to see the authors think more carefully about the impact of TGFb priming on the in vivo osteoclastogenesis and the bone phenotype they have described. In addition, the validity of the observations is compromised by several shortcomings in the methods of investigation, the rigor of the work, and missing critical information. More seriously, the workflow of the entire manuscript is disconnected and complicated to follow. The authors show just a phenomenon in TGFb priming/TNF-driven reprogramming, ISGs, chromatin accessibility, epigenetic changes, IRF1, IRF8, RA but there is no molecular link.

Major concern

1. The authors shows that TGFb priming reprograms osteoclast progenitors response to TNF towards osteoclastogenesis. However, all the in vivo data from their genetic mouse models could not be explained as TGFb priming effects because the deletion of target genes does not mimic the ablation of TGFb priming.
2. The authors consistently use the terms "RANKL-induced physiological osteoclast formation" throughout the manuscript. This is misleading and confusing. As the authors have mentioned, RANKL-induced osteoclastogenesis is important for maintaining normal bone remodeling in physiological condition. However, the role of RANKL has been also well established in pathophysiologic condition including aging, irradiation, and several cancer models. Therefore, RANKL is NOT the only known physiological inducer of osteoclastogenesis.
3. The authors should discuss more details about the role of TGFb signaling in bone homeostasis in the Introduction, instead of summarizing their results.
4. The authors should test the role of TGFb priming in RANKL-induced osteoclast formation system instead of adding OPG in TNF-induced osteoclastogenesis if this effect is indeed in a RANKL independent manner.
5. The micro-CT analysis has not fully achieved in Tgfb2 cKO mice. What about their cortical bone thickness? Did they also analyze their vertebral bones? Why did they choose the young male mice for the experiment? Based on their hypothesis, this reviewer suggests using more inflammatory conditioned mice like aging mice or well-established TNFtg mice.
6. The important control groups, Sham or Vehicle, are missing in the animal experiments (Fig. 2).
7. Most epigenetic approaches have been performed in day 1 condition with TNF even though the authors indicate that "TGFb priming on TNF action in macrophages is a slow biological rather than a rapid cellular response". Did they try to see the chromatin remodeling and transcriptomic changes in later stage of osteoclastogenesis?
8. The authors show that TGF priming affects IRF8 protein stability in the Fig.6, which is not convincing because they consistently show the epigenetic and/or transcriptomic changes in TNF-stimulated osteoclast differentiation throughout the manuscript.
9. RNA-seq analysis (Fig. 7) needs to be done with at least 3 biological replates.

Minor concern

1. The authors should explain exactly what the 'non-priming' condition is. TNF alone? or TNF+TGFb (Macrophage-OC)?
2. The information on gender and age of the several mouse models is missing.
3. The authors need to explain why they examined PBMC-derived macrophages phagocytized zymosan particles and give any conclusion on this approach.
4. Which cell cultures did the author used in Fig. 4?
5. Any bone phenotype from Irf1 KO mice?
6. In the Summary section, is TGFb/TNF-driven macrophage polarization correlated with osteoclastogenesis? This seems to be out of focus in this paper.

Reviewer #2:

Remarks to the Author:

In the manuscript by Xia, Inoue, Zhao, and colleagues, the authors studied the mechanism driving inflammatory osteoclast differentiation. TGF β priming enables TNF to induce osteoclastogenic genes in human CD14+ monocyte-derived macrophages and mouse bone marrow-derived macrophages. TGF β priming modulates TNF action on osteoclastogenic regulators through chromatin remodeling and regulation of histone methylation. TGF β also promotes the degradation of IRF8, an inhibitory transcription factor of osteoclast differentiation. B-Myb induced by TGF β priming/TNF plays an essential role in a TGF β /TNF-driven osteoclastogenic program but dispensable in RANKL-induced osteoclastogenesis in vitro. They also show that TGF β activity positively correlates with osteoclastic gene expression in PBMCs in RA and SLE patients. Overall, the topic is both important and of considerable interest. However, there are several concerns to be clarified that could improve suitability for Nature Communications.

1. TGF β priming enables TNF to induce non-canonical osteoclastogenic regulators, including MYBL2, in vitro. However, it is unclear whether this unique set of osteoclasts exists in vivo. It is essential to show the evidence of B-Myb-high osteoclasts in a pathological model in vivo. It would be even better to show the effect of myeloid cell-specific MYBL2 deletion on inflammatory osteoclastogenesis.
2. TGF β priming/TNF can induce osteoclastogenesis independently of the RANKL pathway in vitro. However, TNF-induced inflammatory osteoclast differentiation is examined under the presence of RANKL in vivo. Does TNF induce inflammatory osteoclastogenesis in RANKL-/- mice?
3. Osteoclast precursors are stimulated with TNF under the presence of TGF β in vivo. However, TGF β is removed before TNF stimulation in vitro. It is unclear whether the removal of TGF β is necessary for the TGF β /TNF-driven osteoclastogenesis.
4. In experiments using the inflammatory calvarial osteolysis mouse model (Figure 2f, 2g, 2h), the data of PBS-injected control mice is necessary to evaluate the effect of myeloid cell-specific Tgfr2 deletion on basal osteoclastogenesis.
5. In Figure 5a, the de novo motif analysis is performed for ATAC-seq peaks in vicinities of the TGF β priming genes and non-priming genes. However, the transcriptional function of those peaks is not assessed. Characterization of H3K27ac, which is highly correlated with regulatory element activity, helps identify transcription factor motifs in accessible chromatin regions with changes in transcriptional activity.
6. In Figure 5a and Figure 7c, the percentage of target and background sequences that have the respective transcription factor motifs should be shown.

Reviewer #3:

Remarks to the Author:

Thank you for offering us the opportunity to review this interesting paper. This study demonstrates TGF β priming+TNF as an alternative molecular pathway for osteoclast generation, which is distinct from the classical RANKL/RANK pathway. The findings are generally interesting and may shed a light on the selective treatment of inflammatory bone destruction.

Major:

1. Please make sure the images in Figure 1a are from the same batch of experiment, as the intensity of TRAP staining in some images (TGF β +, TNF+) is distinct from the others (TGF β -, TNF-). There will be doubt about whether the data is generated from two independent experiments so that they might not be comparable.
2. The study proposed TGF β priming&TNF-induced OCs as inflammatory OCs different from that induced by RANKL under physiological state, then it would be essential to compare the fundamental characteristics (e.g., size/nuclei No. and mineral resorption activity) of these two kinds of OCs.
3. For TGF β priming, the study used 10 ng/mL TGF β for osteoclastic induction of human PBMCs and 1 ng/mL TGF β for osteoclastic induction of murine BMMs. Can you explain why different concentrations of

TGFbeta were used? How does the selected concentration of TGFbeta correlate to the pathological/inflammatory state of RA patients?

4. In Fig.1e the result suggests the effect of TGFbeta on TNF-mediated osteoclastogenesis is independent of RANKL, however, in Fig. 3b the RNAseq data indicates TGFbeta priming enriched RANKL/RANK signaling. Please explain why the results seem controversial.

5. In Fig.1a, the data showed the presence of TGFbeta during Mono-Mφ (priming) prepare the osteoclastic differentiation of macrophage, however, the presence of TGFbeta during Mφ-OC failed to contribute to osteoclastogenesis. What if the TGFbeta is present throughout Mono-Mφ-OC process? This better reflect the pathological situation in RA patients because their TGFbeta level is known to be continuously higher than the healthy individuals.

6. It is reported elsewhere that the addition of TNF-alpha (20 ng/mL) and TGF-beta (10 ng/mL) during OC induction synergistically contribute to the formation of OCs (Quinn JM, Itoh K, Udagawa N, Hausler K, Yasuda H, Shima N, Mizuno A, Higashio K, Takahashi N, Suda T, Martin TJ, Gillespie MT. Transforming growth factor beta affects osteoclast differentiation via direct and indirect actions. *J Bone Miner Res.* 2001 Oct;16(10):1787-94), which seems to contradict to the finding of this study?

7. It is also suggested elsewhere that the addition of TGFbeta and M-CSF (without TNF or RANKL) was sufficient to induce TRAP+ OCs. Moreover, TNF-a Abs failed to suppress osteoclast formation or resorption in M-CSF- and TGFβ-treated cultures (& Itonaga I, Sabokbar A, Sun SG, Kudo O, Danks L, Ferguson D, Fujikawa Y, Athanasou NA. Transforming growth factor-beta induces osteoclast formation in the absence of RANKL. *Bone.* 2004 Jan;34(1):57-64). Please explain.

8. Given the great potential of selective inhibition of inflammatory osteoclasts, it would be both interesting and important to test whether any of the approaches suggested in this study (e.g., inhibition of B-Myb, upregulation of IRF1, or IRF8 degradation inhibition) contribute to selective protection from inflammatory osteoclastogenesis without affecting RANKL-mediated osteoclastogenesis in an animal model.

9. TGFbeta has been known to downregulate RANKL while upregulating OPG from OBs, which suppresses RANKL-driven osteoclastogenesis. Would the TGFbeta priming+TNF-driven osteoclastogenesis override the osteogenic effect of TGFbeta?

Minor:

1. All the quantification data should be presented in dot plot to show the data distribution and number of biological replicates (e.g. 1a, 1c, 1e, 1g, 2a, 2d, 2e...)

2. Please clearly indicate (especially in the result and method part) whether the TNF refers to TNF-alpha. Addition, which isoform(s) of TGF-beta was/were used in this study?

Response to Reviewers

NCOMMS-21-20341-A: "TGF β reprograms TNF stimulation of macrophages towards a non-canonical pathway driving inflammatory osteoclastogenesis" by Yuhan Xia, Inoue Kazuki, Yong Du, Stacey J. Baker, E. Premkumar Reddy, Matthew B. Greenblatt and Baohong Zhao.

We thank the reviewers for their time and their positive and insightful comments. We are pleased that the reviewers were very enthusiastic: "*interesting paper*"; "*a novel function of TGF β* "; "*data is solid*"; "*the topic is both important and of considerable interest*"; "*findings are interesting*"; "*may shed a light on the selective treatment of inflammatory bone destruction*". We have made every effort to overcome the difficulties from the COVID-19 pandemic, and have experimentally addressed all of the points raised by the reviewers and generated 66 new figure panels of data. We have revised the manuscript accordingly and the reviewers' points are specifically addressed below. Changes in the manuscript have been underlined.

Response to specific points:

Reviewer #1 (Remarks to the Author):

Xia and colleagues have studied the role of TGF β /TNF-driven inflammatory osteoclastogenic program in excessive bone resorption condition. This interesting paper shows a novel function of TGF β in the regulation of TNF action on osteoclast differentiation via RANKL-independent pathway. The data is solid, but I would like to see the authors think more carefully about the impact of TGF β priming on the in vivo osteoclastogenesis and the bone phenotype they have described. In addition, the validity of the observations is compromised by several shortcomings in the methods of investigation, the rigor of the work, and missing critical information. More seriously, the workflow of the entire manuscript is disconnected and complicated to follow. The authors show just a phenomenon in TGF β priming/TNF-driven reprogramming, ISGs, chromatin accessibility, epigenetic changes, IRF1, IRF8, RA but there is no molecular link.

We thank the reviewer's comments, have addressed the questions in response to the major and minor concerns, and also appreciate this opportunity to explain the question mentioned in the last sentence: molecular link between the phenotypes and underlying molecular mechanisms, more clearly. The TGF β priming/TNF-driven reprogramming program is primarily reflected by transcriptome changes, featured by changes in ISG expression and osteoclastic gene expression. Because epigenetic changes, such as chromatin accessibility and histone modifications, often change dynamically in response to environmental cues and play fundamental roles in the regulation of gene expression, we then investigated the epigenetic changes induced by TGF β priming and indeed found that TGF β priming remodeled chromatin accessibility and histone modifications of ISGs and osteoclastic genes in response to TNF. Through these actions, TGF β creates a favorable chromatin environment for TNF to induce macrophages to differentiate into osteoclasts, instead of polarizing to inflammatory macrophages. In addition to the chromatin remodeling, TGF β priming enables TNF to induce a novel non-canonical osteoclastogenic program, which includes the suppression of the TNF-induced IRF1-IFN β -IFN-stimulated-gene axis, IRF8 degradation and B-Myb induction, to join the concert to drive macrophages to differentiate into osteoclasts. These mechanisms are non-exclusively integrated into the novel non-canonical osteoclastogenic program to regulate and

coordinate TNF-induced inflammatory osteoclastogenesis together. Behind a biological phenomenon, there are always multiple mechanisms. For example, it is well established that RANKL-induced osteoclast differentiation is regulated by many mechanisms, including various transcription factors, signaling pathways and epigenetic regulation. We appreciate the cutting-edge sequencing techniques we applied in this study, which enabled us to identify a novel differentiation program that includes multiple important mechanisms for TGF β priming/TNF-driven osteoclastogenesis. Moreover, these molecular mechanisms are active in rheumatoid arthritis, in which TGF β level is elevated and correlates with osteoclast activity. This link between these mechanisms and RA thus indicates the importance and clinical relevance of the identified mechanisms in a disease setting. Most of these points have been discussed in the Discussion section, especially the second paragraph on pg. 12 and the last two paragraphs on pg. 14. In addition, the introduction and summary descriptions at the beginning and the end of each relevant result section help explain the links.

Major concern

1. The authors shows that TGF β priming reprograms osteoclast progenitors response to TNF towards osteoclastogenesis. However, all the in vivo data from their genetic mouse models could not be explained as TGF β priming effects because the deletion of target genes does not mimic the ablation of TGF β priming.

We thank the reviewer for this question. In this manuscript, we generated TGF β receptor 2 conditional knock out (KO) mice, in which TGF β receptor 2 is specifically deleted in myeloid lineage macrophages/osteoclast precursors by crossing *Tgfb2^{fllox/fllox}* mice with *LysMcre* mice (*Tgfb2^{flf};LysMCre*; hereafter referred to as *Tgfb2 Δ^M*). Sex-matched *LysMcre⁺* littermates served as wild type (WT) controls. There is a basal level of TGF β in the physiological condition in WT mice, which is thought to correspond to a TGF β priming condition *in vivo*. The macrophages in *Tgfb2 Δ^M* mice lack basal TGF β signaling because of TGF β receptor 2 deletion, thus mimicking the TGF β non-priming condition *in vivo*. Indeed, our experiments show that there exists basal expression of the target genes of TGF β signaling, indicating a basal level of TGF β that can mimic TGF β priming, in the macrophages from WT mice. In contrast, the expression of TGF β signaling target genes is drastically diminished in *Tgfb2 Δ^M* macrophages. These new results are now shown in Suppl. Fig. 7b. Meanwhile, we understand the reviewer's concern and are also aware of the difficulty of establishing an identical in vivo model as that in in vitro. In fact, however, it is almost always challengeable to find an in vivo system that perfectly matches in vitro conditions in biological field. Therefore, we wished to take advantage of these mouse models (*Tgfb2 Δ^M* mimics non-priming, and WT mimics TGF β priming) to **provide a proof of concept** for TGF β priming effect on osteoclasts and bone resorption *in vivo*. Using these mouse models is the most feasible, readily available, characteristically close to in vitro system, and reliable genetic approach to study TGF β priming effect in vivo and fits the purpose of this study. Along the advance of technology, we hope there might appear a perfect in vivo model in future, which may take considerable time and is beyond the time frame of this revision. The new results are shown in Suppl. Fig. 7b. Relevant points are discussed on pg. 5.

2. The authors consistently use the terms "RANKL-induced physiological osteoclast formation" throughout the manuscript. This is misleading and confusing. As the authors have mentioned, RANKL-induced osteoclastogenesis is important for maintaining normal bone remodeling in physiological condition. However, the role of RANKL has been also well established in

pathophysiologic condition including aging, irradiation, and several cancer models. Therefore, RANKL is NOT the only known physiological inducer of osteoclastogenesis.

We agree with the reviewer's comment and have deleted the word 'physiological' from 'RANKL-induced physiological osteoclast formation' throughout the manuscript.

3. The authors should discuss more details about the role of TGF β signaling in bone homeostasis in the Introduction, instead of summarizing their results.

Following the reviewer's suggestion, we have added more discussion on the role of TGF β signaling in bone homeostasis and RANKL-induced osteoclastogenesis in the Introduction on pg. 3.

4. The authors should test the role of TGF β priming in RANKL-induced osteoclast formation system instead of adding OPG in TNF-induced osteoclastogenesis if this effect is indeed in a RANKL independent manner.

Following the reviewer's suggestion, we have tested the role of TGF β priming in RANKL-induced osteoclast formation. The results show that TGF β priming does not affect RANKL-induced osteoclast formation. The data is now shown in Suppl. Fig. 6b and noted in the text on pp. 5 and 29.

*5. The micro-CT analysis has not fully achieved in *Tgfb2* cKO mice. What about their cortical bone thickness? Did they also analyze their vertebral bones? Why did they choose the young male mice for the experiment? Based on their hypothesis, this reviewer suggests using more inflammatory conditioned mice like aging mice or well-established TNFtg mice.*

Following the reviewer's suggestion, we have performed additional μ CT scans and analyzed the cortical bone phenotype as well as the vertebral bone phenotype of the mice. As shown in Suppl. Fig. 8, *Tgfb2* deficiency does not affect cortical bone thickness (Suppl. Fig. 8a) or vertebral bone phenotype (Suppl. Fig. 8b). In this study, we first examined the bone phenotype in 12-week old mice because it is essential to understand the effect of TGF β signaling in myeloid cells on the basal physiological bone phenotype in bone homeostasis (Fig. 2b, c). 12-week old mice are generally considered to enter bone homeostatic stage from growth phase. The data indicate that TGF β signaling plays a dispensable role in osteoclastogenesis under physiological conditions. We next investigated whether lack of TGF β receptor 2 impacts osteoclastogenesis and bone resorption in inflammatory conditions and have applied two inflammatory animal models; a well-established inflammatory calvarial osteolysis mouse model induced by TNF (Fig. 2f, g, h), and the K/BxN serum-induced arthritis model (Fig. 2i, j, k). The results show that TGF β signaling plays a crucial role in promoting TNF-mediated osteoclastogenesis and exacerbating inflammatory bone resorption in these inflammatory models. Aging includes many physiopathological changes in addition to inflammation, thus aging model is not appropriate for the purpose of this study. Moreover, aging process takes at least 1-2 years to establish the models, which is quite beyond the time frame of this revision. The reviewer's suggestion of using TNFtg mice will need crossing TNFtg mice with *Tgfb2f/f;LysMCre* mice to generate *TNFtg;Tgfb2f/f;LysMCre* mice that have three allele mutations. With appropriate littermate controls needed, it will take more than 2 years to gather enough numbers of the mouse pairs, which is not feasible for the time frame of the revision, and particularly difficult in the Covid pandemic. However, the reviewer's comment is taken, and we

have spent our full efforts in this revision to establish a long-term TNF-induced inflammatory bone resorption model, in addition to the two inflammatory mouse models we have already applied. In this long-term TNF-induced inflammatory bone resorption model, recombinant TNF was injected to the calvarial periosteum daily for fourteen days to mimic a chronic TNF-induced inflammatory conditions in vivo. Because this is a TNF-induced inflammatory bone resorption model, it fits our study purpose very well. Compared to the PBS injection groups as the control, long-term TNF treatment induces not only bone resorption on the calvarial bones (Supplementary Fig. 9a), but also significant trabecular bone loss in femurs and vertebrae in the WT mice (Supplementary Fig. 9b, c). In a striking contrast, lack of TGF β signaling protects bone loss in calvarial, femoral and vertebral bones induced by TNF in the *Tgfb2* ^{Δ M} mice (Supplementary Fig. 9a, b, c). These results furthermore demonstrate that TGF β signaling plays a critical role in enhancing TNF-mediated inflammatory bone loss. The data is now shown in Suppl. Fig. 9 and noted in the text on pp. 6 and 30.

6. The important control groups, Sham or Vehicle, are missing in the animal experiments (Fig. 2).

The PBS injection groups as the control are now added to the animal experiments in Fig. 2f-k. The results show that myeloid specific deletion of *Tgfb2* does not affect basal osteoclastogenesis in the control groups. The new results are now shown in Fig. 2f-k, and noted in the text on pp. 6 and 24.

7. Most epigenetic approaches have been performed in day 1 condition with TNF even though the authors indicate that “TGF β priming on TNF action in macrophages is a slow biological rather than a rapid cellular response”. Did they try to see the chromatin remodeling and transcriptomic changes in later stage of osteoclastogenesis?

Thanks for the reviewer’s comment. We did not explain clearly what a rapid cellular response is like in the mentioned sentence. We now provide an example for a rapid cellular response in that sentence: “TGF β priming on TNF action in macrophages is a slow biological rather than a rapid cellular response, such as a rapid immune response to toll-like receptor stimulation occurring within minutes to hours...” on pg. 14. One day reaction is regarded slow when compared to a rapid cellular response occurring within minutes to hours. We hope this explanation can make this sentence clearer.

Following the reviewer’s question, we furthermore performed FAIRE (formaldehyde-assisted isolation of regulatory element)-qPCR assay to quantify chromatin compaction/accessibility at the ISG and OC gene loci during chromatin remodeling at late stage (d6) of osteoclastogenesis. The results show that TGF β priming continuously drastically enhances the chromatin accessibility at the loci of OC genes, such as NFATC1, ITGB3, ACP5, CALCR and MYBL2, at day 6 of TNF stimulation of macrophages (Supplementary Fig. 12a). Fos locus is an exception in that its chromatin became closed at day 6 regardless of TGF β priming or not (Supplementary Fig. 12a). In contrast, most ISG gene loci, such as MX1, MX2, IFIT2, IFIT3, CXCL9, CXCL10, CXCL11, IFNB1 and IRF1, at day 6 of TNF treatment turn into closed state, which is not affected or further suppressed by TGF β priming (Supplementary Fig. 12b). These chromatin remodeling changes at late stage of TNF stimulation of macrophages are consistent with the transcriptomic results of OC genes and ISG genes shown in Fig. 3c-e, in which ISG gene expression (Fig. 3c, d) is almost completely suppressed at day 6 of TNF stimulation, while the

expression of OC genes, except for FOS, is highly induced day 6 of TNF stimulation in the TGF β priming condition (Fig. 3e). These results are now shown in Fig. 3c-e and Supplementary Fig. 12, and noted in the text on pp. 7, 8, 19 and 30.

8. The authors show that TGF priming affects IRF8 protein stability in the Fig.6, which is not convincing because they consistently show the epigenetic and/or transcriptomic changes in TNF-stimulated osteoclast differentiation throughout the manuscript.

As we explained in response to the reviewer's summary comments on page 1, TGF β regulated IRF8 degradation is one of the mechanisms that belong to the newly identified non-canonical osteoclastogenic program. This mechanism mediated by IRF8 protein stability is not an epigenetic or transcriptomic change, but is an important mechanism for TGF β priming/TNF-driven osteoclastogenesis.

9. RNA-seq analysis (Fig. 7) needs to be done with at least 3 biological replates.

Following the reviewer's suggestion, we performed RNA-seq analysis with 3 biological replicates for both RANKL (GSE171542) and TGF β priming/TNF (GSE171843) conditions in Fig. 7. The results show that using 3 biological replicates has improved the statistical significance, which is indicated by the adjusted p-value in Fig. 7b (Y axis), of DEGs without impacting conclusion. We here thank the reviewer's suggestion, which has further strengthened our conclusion. The results are shown in Fig. 7a, b and noted on pp. 18, 27.

Minor concern

1. The authors should explain exactly what the 'non-priming' condition is.

TNF alone? or TNF+TGF β (Macrophage-OC)?

The "non-priming condition" is the TNF alone condition. We have now added an explanation "...TNF alone (hereafter referred to as non-priming) ..." in the Results section on pg. 5, and noted on the schematic in Suppl. Fig. 1.

2. The information on gender and age of the several mouse models is missing.

We added the information on gender and age of the mouse models in the figure legends of Fig. 2, Fig. 7, Supplemental Fig. 8, and Supplemental Fig. 9.

3. The authors need to explain why they examined PBMC-derived macrophages phagocytized zymosan particles and give any conclusion on this approach.

Phagocytosis is an important function of macrophages. Examination of phagocytosis of zymosan particles is a well-established approach evaluating the phagocytic function of macrophages. In this study, we examined whether TGF β priming affects macrophage phagocytic function. As shown in Fig. 1f, PBMC-derived macrophages phagocytized zymosan particles as expected. However, TGF β priming almost completely abolished the phagocytic characteristics of macrophages (Fig. 1f). These results are discussed on pg. 5.

4. Which cell cultures did the author used in Fig. 4?

Human CD14(+) monocytes treated with or without TGF β priming for 3 days, followed by TNF stimulation for 0 or 1 day were used. This information has now been added to the Fig.4 legend on pg. 25.

5. Any bone phenotype from Irf1 KO mice?

The Irf1 bone phenotype has been reported in the literature. Both global (*Irf1*^{-/-}) and myeloid specific (*Irf1*^{fl/fl};LysMCre) Irf1 knockout mice show osteoporotic phenotypes with reduced BV/TV and increased trabecular spacing. Also, Irf1 deficiency enhances osteoclast differentiation *in vitro* (Place, D.E., et al. *Nat Commun.* 12, 496 (2021). PMID: 33479228).

6. In the Summary section, is TGF β /TNF-driven macrophage polarization correlated with osteoclastogenesis? This seems to be out of focus in this paper.

The osteoclastogenesis is a type of macrophage polarization, from a broad view or immunological point of view that refers to cell reaction to stimuli. However, the reviewer's comment is taken, and we changed the wording to macrophage polarization/differentiation in the Summary section on pg. 14.

Reviewer #2 (Remarks to the Author):

*In the manuscript by Xia, Inoue, Zhao, and colleagues, the authors studied the mechanism driving inflammatory osteoclast differentiation. TGF β priming enables TNF to induce osteoclastogenic genes in human CD14+ monocyte-derived macrophages and mouse bone marrow-derived macrophages. TGF β priming modulates TNF action on osteoclastogenic regulators through chromatin remodeling and regulation of histone methylation. TGF β also promotes the degradation of IRF8, an inhibitory transcription factor of osteoclast differentiation. B-Myb induced by TGF β priming/TNF plays an essential role in a TGF β /TNF-driven osteoclastogenic program but dispensable in RANKL-induced osteoclastogenesis *in vitro*. They also show that TGF β activity positively correlates with osteoclastic gene expression in PBMCs in RA and SLE patients. Overall, the topic is both important and of considerable interest. However, there are several concerns to be clarified that could improve suitability for Nature Communications.*

*1. TGF β priming enables TNF to induce non-canonical osteoclastogenic regulators, including MYBL2, *in vitro*. However, it is unclear whether this unique set of osteoclasts exists *in vivo*. It is essential to show the evidence of B-Myb-high osteoclasts in a pathological model *in vivo*. It would be even better to show the effect of myeloid cell-specific MYBL2 deletion on inflammatory osteoclastogenesis.*

Following the reviewer's suggestion, we performed immunohistochemical staining of B-Myb on the slices of calvarial bones obtained from mice in the TNF-induced supracalvarial osteolysis mouse model. The expression of B-Myb was clearly identified in osteoclasts *in vivo* in the TNF-induced inflammatory osteolysis model (Fig. 7m). The results are shown in Fig. 7m, and noted on pp. 10, 19 and 27.

Following the reviewer's suggestion, we further generated myeloid specific *Mybl2* conditional KO mice (*Mybl2*^{fl/fl};LysMCre; hereafter referred to as *Mybl2* ^{Δ^M}) by crossing *Mybl2*^{fl/fl} mice with

LysMCre mice, and examined the effects of myeloid cell-specific *Mybl2* deletion on TNF-induced inflammatory osteoclastogenesis and bone resorption *in vivo*. Compared to the WT control mice, *Mybl2* deficiency significantly suppressed TNF-induced calvarial bone erosions (Fig. 7n), and markedly decreased osteoclast formation in the calvarial bones from the *Mybl2*^{ΔM} mice (Fig. 7o, p, TNF group). Myeloid cell-specific *Mybl2* deletion did not affect osteoclast formation under physiological condition (Fig. 7o, p, PBS group). The results are shown in Fig. 7n, o, p, and noted on pp. 10, 15 and 27.

We here thank the reviewer for these important suggestions, the experiments and results based on which have further highlighted the significance and novelty of our study.

2. TGF?? priming/TNF can induce osteoclastogenesis independently of the RANKL pathway in vitro. However, TNF-induced inflammatory osteoclast differentiation is examined under the presence of RANKL in vivo. Does TNF induce inflammatory osteoclastogenesis in RANKL^{-/-} mice?

Papadaki M, *et al.* reported that TNF can induce RANKL-independent osteoclastogenesis in the inflamed ankle joints. They crossed human TNF α -Tg mice (Tg197 mice) with *Rankl*^{fl^{es}/fl^{es}} mice. The *Rankl*^{fl^{es}/fl^{es}} mice carry a functional mutation in the RANKL gene, which inhibits RANKL trimerization and binding to RANK, and develop osteopetrotic bone phenotype similarly to *RANKL*^{-/-} mice. TRAP+ osteoclasts were identified in the inflamed joints of Tg197/*Rankl*^{fl^{es}/fl^{es}} mice. (Papadaki M, *et al. Front Immunol.* 2019 Feb 5;10:97. PMID: 30804932)

In addition, TRAP+ and cathepsin K+ osteoclasts were identified on calvarial bone surfaces of the *RANKL*^{-/-} mice in TNF-induced calvarial osteolysis model. (Li J, *et al. Proc Natl Acad Sci U S A.* 2000 Feb 15; 97(4): 1566–1571. PMID: 10677500)

These studies showed that TNF induces inflammatory osteoclastogenesis independent of RANK/RANKL signaling *in vivo*.

3. Osteoclast precursors are stimulated with TNF under the presence of TGF?? in vivo. However, TGF?? is removed before TNF stimulation in vitro. It is unclear whether the removal of TGF?? is necessary for the TGF??/TNF-driven osteoclastogenesis.

Following the reviewer's question, we maintained TGF β in our culture system without removal to mimic *in vivo*, and found that TGF β together with TNF induced a lot of giant TRAP+ osteoclasts while TNF alone did not (Suppl. Fig. 3). These results indicate that the removal of TGF β is not necessary for the TGF β /TNF-driven osteoclastogenesis. The results are shown in Suppl. Fig. 3, and noted on pp. 5 and 29.

4. In experiments using the inflammatory calvarial osteolysis mouse model (Figure 2f, 2g, 2h), the data of PBS-injected control mice is necessary to evaluate the effect of myeloid cell-specific Tgfb2 deletion on basal osteoclastogenesis.

The PBS injection groups as the control are now added to the animal experiments in Fig. 2f-k. The results show that myeloid specific deletion of *Tgfb2* does not affect basal osteoclastogenesis in the control groups. The new results are now shown in Fig. 2f-k, and noted in the text on pp. 6 and 24.

5. In Figure 5a, the de novo motif analysis is performed for ATAC-seq peaks in vicinities of the TGFβ priming genes and non-priming genes. However, the transcriptional function of those peaks is not assessed. Characterization of H3K27ac, which is highly correlated with regulatory element activity, helps identify transcription factor motifs in accessible chromatin regions with changes in transcriptional activity.

Following the reviewer's suggestion, we performed Cut&Run-seq for the H3K27ac histone mark to assess the transcriptional function of the ATAC peaks we identified. Indeed, we found that H3K27ac signals were detected around the ATAC peaks. At non-priming gene loci, including ISGs, H3K27ac signals were present at baseline and increased after TNF stimulation (Fig. 4j, k, l, m). These H3K27ac signals were drastically attenuated by TGFβ priming (Fig. 4j, k, l, m). On the contrary, both basal and TNF-stimulated levels of H3K27ac signals were low at priming gene loci, such as OC genes, but TGFβ priming strikingly elevated these H3K27ac signals with even higher levels with TNF stimulation (Fig. 4j, k, l, n, and Supplementary Fig. 11). These H3K27ac signal changes are consistent with the H3K4me3 mark and transcriptomic changes of ISGs and OC genes. These new results are now shown in Fig. 4j, k, l, m, n, and Supplementary Fig. 11, and discussed on pp. 8 and 26.

6. In Figure 5a and Figure 7c, the percentage of target and background sequences that have the respective transcription factor motifs should be shown.

Following the reviewer's suggestion, we added the percentage of target and background sequences in Fig. 5a and Fig. 7c.

Reviewer #3 (Remarks to the Author):

Thank you for offering us the opportunity to review this interesting paper. This study demonstrates TGFβ priming+TNF as an alternative molecular pathway for osteoclast generation, which is distinct from the classical RANKL/RANK pathway. The findings are generally interesting and may shed a light on the selective treatment of inflammatory bone destruction.

Major:

1. Please make sure the images in Figure 1a are from the same batch of experiment, as the intensity of TRAP staining in some images (TGFβ+, TNF+) is distinct from the others (TGFβ-, TNF-). There will be doubt about whether the data is generated from two independent experiments so that they might not be comparable.

We thank the reviewer for this comment. These images are indeed from the same experiment, but probably due to the problem of microscopic setting, the background color showed differently between some images. We have reset the microscope, retaken the photos and replaced the old images. Now the images exhibit same intensity of TRAP staining among conditions in Fig. 1a. In addition, the experiment in Fig. 1a has been repeated at least 5 times with quantification data shown in the right panel of Fig. 1a.

2. The study proposed TGFβ priming&TNF-induced OCs as inflammatory OCs different from that induced by RANKL under physiological state, then it would be essential to compare the

fundamental characteristics (e.g., size/nuclei No. and mineral resorption activity) of these two kinds of OCs.

We compared the number of nuclei, the size and resorption activity of osteoclasts induced by RANKL or TGF β priming/TNF. As shown in the Supplemental Figure 4, the fundamental characteristics of these osteoclasts are similar, further demonstrating that TGF β priming/TNF has strong capacity to induce mature osteoclasts as those induced by RANKL. Notably, this is not contradictory to our conclusion that TGF β priming/TNF induces novel non-canonical osteoclastogenic mechanisms, including epigenetic regulation, the suppression of the TNF-induced IRF1-IFN β -IFN-stimulated-gene axis, IRF8 degradation and B-Myb induction. These TGF β priming/TNF-induced osteoclastogenic mechanisms (not osteoclasts) are different from those induced by RANKL. These distinct osteoclastogenic mechanisms open avenues for selective treatment of inflammatory osteolysis. The new results are shown in Supplemental Figure 4, and noted on pp. 5 and 29.

3. For TGFbeta priming, the study used 10 ng/mL TGFbeta for osteoclastic induction of human PBMCs and 1 ng/mL TGFbeta for osteoclastic induction of murine BMMs. Can you explain why different concentrations of TGFbeta were used? How does the selected concentration of TGFbeta correlate to the pathological/inflammatory state of RA patients?

TGF β priming has a dose-dependent effect on TNF mediated osteoclast differentiation (Supplementary Fig. 2). We chose the concentrations of TGF β used in this study based on their priming efficiency. For human CD14+ cells, 10 ng/ml of TGF β is able to effectively induce multinucleated osteoclasts (Supplemental Figure 2a). For mouse BMM cells, 1 ng/ml of TGF β is sufficient to induce osteoclastogenesis (Supplemental Figure 2b). The cells from different species often show different dose-dependent curves in response to certain stimulation, such as the human and mouse cell response to TGF β in our study. The new results are shown in Supplementary Fig. 2, and noted on pp. 5 and 29.

4. In Fig. 1e the result suggests the effect of TGFbeta on TNF-mediated osteoclastogenesis is independent of RANKL, however, in Fig. 3b the RNAseq data indicates TGFbeta priming enriched RANKL/RANK signaling. Please explain why the results seem controversial.

There are a large body of studies focusing on RANKL-induced osteoclast differentiation in the past 2 decades. Based on these studies, some databases, such as WikiPathways used for the pathway analysis of RNAseq data, utilize 'RANKL/RANK signaling' pathway to name canonical osteoclast pathway. The 'RANKL/RANK signaling' pathway in WikiPathways include genes not only involved in RANK/RANKL signaling but also genes for canonical osteoclastogenesis. However, our results show that TGF β priming/TNF-induced osteoclastogenesis is independent of RANKL, supported by not only data in Suppl. Fig. 6 but also the results that RANK expression was not affected by TGF β -priming shown as the below figure. Thus, it was the pathway name 'RANKL/RANK signaling' from WikiPathways used in Fig. 3b that is misleading and does not reflect our results. We thank the reviewer for pointing this out to let us prevent confusion. However, we cannot change that pathway name, which was created by each database. We thus used another well-established database, KEGG, to reanalyze our RNAseq data and found that 'osteoclast differentiation' pathway in KEGG was significantly enriched by TGF β . We have replaced the data accordingly in Fig. 3b, which does not affect our conclusion.

5. In Fig. 1a, the data showed the presence of TGFbeta during Mono-M?? (priming) prepare the osteoclastic differentiation of macrophage, however, the presence of TGFbeta during M??-OC failed to contribute to osteoclastogenesis. What if the TGFbeta is present throughout Mono-M??-OC process? This better reflect the pathological situation in RA patients because their TGFbeta level is known to be continuously higher than the healthy individuals.

Following the reviewer's question, we maintained TGFβ in our culture system without removal to mimic in vivo, and found that TGFβ together with TNF induced a lot of giant TRAP+ osteoclasts while TNF alone did not (Suppl. Fig. 3). These results indicate that the removal of TGFβ is not necessary for the TGFβ/TNF-driven osteoclastogenesis. The results are shown in Suppl. Fig. 3, and noted on pp. 5 and 29.

6. It is reported elsewhere that the addition of TNF-alpha (20 ng/mL) and TGF-beta (10 ng/mL) during OC induction synergistically contribute to the formation of OCs (Quinn JM, Itoh K, Udagawa N, Hausler K, Yasuda H, Shima N, Mizuno A, Higashio K, Takahashi N, Suda T, Martin TJ, Gillespie MT. Transforming growth factor beta affects osteoclast differentiation via direct and indirect actions. *J Bone Miner Res.* 2001 Oct;16(10):1787-94), which seems to contradict to the finding of this study?

In this paper, they cultured M-CSF-stimulated mouse spleen cells or RAW264.7 cells with TGFβ and TNFα for 7 days. M-CSF-stimulated spleen cells with TGFβ and TNFα during OC induction induced only a few osteoclasts (< 20 TRAP+ MNCs per well). In our studies, TGFβ priming enables TNF to induce more than 200 large TRAP+ MNCs per well. Furthermore, spleen cell cultures or RAW 264.7 cell cultures do not always recapitulate primary monocytes/macrophage cultures, and are especially different from human monocyte cultures. Because the cells and culture systems are quite different, it is difficult to make conclusive comparison between the findings of this paper and our study.

7. It is also suggested elsewhere that the addition of TGFbeta and M-CSF (without TNF or RANKL) was sufficient to induce TRAP+ OCs. Moreover, TNF-a Abs failed to suppress osteoclast formation or resorption in M-CSF- and TGF??-treated cultures (& Itonaga I, Sabokbar A, Sun SG, Kudo O, Danks L, Ferguson D, Fujikawa Y, Athanasou NA. Transforming growth factor-beta induces osteoclast formation in the absence of RANKL. *Bone.* 2004 Jan;34(1):57-64). Please explain.

In this paper, M-CSF and RANKL were able to induce numerous large (50–150 μm diameter) TRAP+ multinuclear osteoclasts. However, most of the M-CSF and TGF β -induced osteoclasts (>75%) were mononuclear TRAP+ cells and were also small (<50 μm diameter) containing fewer than five nuclei. Also, it took 14-21 days for M-CSF and TGF β to induce osteoclastogenesis, which is much longer than our culture system. In our system, TGF β -priming/TNF induced osteoclasts are comparable to RANKL-induced osteoclasts (Supplemental Figure 4). However, in their culture system, the number of M-CSF and TGF β -induced osteoclasts was significantly lower than that in M-CSF + RANKL osteoclasts (5 ± 0.99 vs 23 ± 10.2 cells). The resorption activity of M-CSF and TGF β -induced osteoclasts was 10 times lower than RANKL-induced osteoclasts ($2.25 \pm 0.99\%$ vs $27.9 \pm 10.5\%$). Thus, the capacity and efficiency of M-CSF and TGF β -induced osteoclastogenesis are much weaker than those of TGF β -priming/TNF-induced osteoclastogenesis as shown in our study.

They used TNF α Ab to block **endogenous** TNF α in their culture, not exogenous TNF α . Endogenous TNF α levels in culture medium of monocytes/macrophages are normally very low (< 10 pg/mL, Nguyen J. et al. FEBS Lett. 2005 Oct 24;579(25):5487-93. doi:10.1016/j.febslet.2005.09.012. PMID: 16213498). Whether M-CSF and TGF β -induced osteoclastogenesis is affected by exogenous TNF at various doses was not tested in that paper.

8. Given the great potential of selective inhibition of inflammatory osteoclasts, it would be both interesting and important to test whether any of the approaches suggested in this study (e.g., inhibition of B-Myb, upregulation of IRF1, or IRF8 degradation inhibition) contribute to selective protection from inflammatory osteoclastogenesis without affecting RANKL-mediated osteoclastogenesis in an animal model.

Following the reviewer's suggestion, we further generated myeloid specific *Mybl2* conditional KO mice (*Mybl2^{fl/fl};LysMCre*; hereafter referred to as *Mybl2 Δ^M*) by crossing *Mybl2^{fl/fl}* mice with *LysMCre* mice, and examined the effects of myeloid cell-specific *Mybl2* deletion on TNF-induced inflammatory osteoclastogenesis and bone resorption *in vivo*. Compared to the WT control mice, *Mybl2* deficiency significantly suppressed TNF-induced calvarial bone erosions (Fig. 7n), and markedly decreased osteoclast formation in the calvarial bones from the *Mybl2 Δ^M* mice (Fig. 7o, p, TNF group). Myeloid cell-specific *Mybl2* deletion did not affect osteoclast formation under physiological condition (Fig. 7o, p, PBS group). The results are shown in Fig. 7n, o, p, and noted on pp. 10, 15 and 27.

We here thank the reviewer for this important suggestion, the experiments and results based on which have further highlighted the significance and novelty of our study.

9. TGFbeta has been known to downregulate RANKL while upregulating OPG from OBs, which suppresses RANKL-driven osteoclastogenesis. Would the TGFbeta priming+TNF-driven osteoclastogenesis override the osteogenic effect of TGFbeta?

Literature (ref1 and ref2 listed as below) provided evidence that TNF can induce osteoclastogenesis in the absence of RANKL/RANK signaling. In these mice, TGF β is present (meaning that TGF β mediated osteogenic effect is present) but RANKL/RANK signaling is absent (meaning no RANKL-induced osteoclastogenesis in these mice), which mimic the condition brought out by the reviewer. In this condition, TNF treatment with pre-existing TGF β (TGF β is present in these mice), which mimics TGF β priming/TNF stimulation, can still induce

osteoclastogenesis and bone resorption. These results indicate that TGF β priming/TNF-driven osteoclastogenesis is highly likely able to override the osteogenic effect of endogenous TGF β . We meanwhile understand the complexity of these cross-talks/cross-regulations happening in vivo, in particular with different doses of TGF β and TNF as well as in different mouse models testing both osteogenesis and bone resorption. A thorough and detailed investigation on this question is beyond the scope of the current study but would be expected in future studies.

Ref1: Papadaki M, *et al.* reported that TNF can induce RANKL-independent osteoclastogenesis in the inflamed ankle joints. They crossed human TNF α -Tg mice (Tg197 mice) with *Rankl*^{fl^{es}/fl^{es}} mice. The *Rankl*^{fl^{es}/fl^{es}} mice carry a functional mutation in the RANKL gene, which inhibits RANKL trimerization and binding to RANK, and develop osteopetrotic bone phenotype similarly to *RANKL*^{-/-} mice. TRAP+ osteoclasts were identified in the inflamed joints of Tg197/*Rankl*^{fl^{es}/fl^{es}} mice. (Papadaki M, *et al. Front Immunol.* 2019 Feb 5;10:97. PMID: 30804932)

Ref2: TRAP+ and cathepsin K+ osteoclasts were identified on calvarial bone surfaces of the *RANK*^{-/-} mice in TNF-induced calvarial osteolysis model. (Li J, *et al. Proc Natl Acad Sci U S A.* 2000 Feb 15; 97(4): 1566–1571. PMID: 10677500)

Minor:

1. All the quantification data should be presented in dot plot to show the data distribution and number of biological replicates (e.g. 1a, 1c, 1e, 1g, 2a, 2d, 2e???)

Following the reviewer's suggestion, we changed all bar graphs to bar graphs with dot plots.

2. Please clearly indicate (especially in the result and method part) whether the TNF refers to TNF-alpha. Addition, which isoform(s) of TGF-beta was/were used in this study?

Thanks for these points. We indicated that TNF refers to TNF α in the result and method part. We used TGF β 1 and also indicated it in the results and method part.

We feel that the revised manuscript addresses the issues raised during review and the manuscript has been significantly strengthened. Thank you for considering the revised manuscript for publication in *Nature Communications*.

REVIEWERS' COMMENTS

Reviewer #1 (Remarks to the Author):

The authors well addressed to this reviewer's comments and suggestions. Therefore, I believe the manuscript has improved much better than the previous version and it is certainly publishable in Nature Communications because of high impact and novelty of this study.

Reviewer #2 (Remarks to the Author):

The authors have addressed my main concerns. The newly provided data further strength their main conclusions. I have no further comments.

Reviewer #3 (Remarks to the Author):

In the revised version of their article and the rebuttal letter, the authors have addressed most of the questions and comments from the reviewer, however, we would like suggest the following:

1. To prevent confusion, the author chose KEGG instead of WikiPathways database for in vitro study using primary human macrophages. But for analysis of Gene Expression in PBMCs from RA and SLE patients, the author keeps using WikiPathways database. Justification for the selection of different database needs to be included in the discussion.
2. The authors used LysMCre+ littermates as control. However, it was noted as WT in the figures and many other parts of the manuscript, which is incorrect. The control used need to be cleared properly noted.

Response to Reviewers

NCOMMS-21-20341-B: "TGF β reprograms TNF stimulation of macrophages towards a non-canonical pathway driving inflammatory osteoclastogenesis" by Yuhan Xia, Inoue Kazuki, Yong Du, Stacey J. Baker, E. Premkumar Reddy, Matthew B. Greenblatt and Baohong Zhao.

We thank the reviewers for their time, and are pleased that Reviewers 1 and 2 are satisfied with our revision. We have addressed the points raised by Reviewer 3 as below and revised the manuscript accordingly. Changes in the manuscript have been underlined.

Response to specific points:

Reviewer #3 (Remarks to the Author): In the revised version of their article and the rebuttal letter, the authors have addressed most of the questions and comments from the reviewer, however, we would like suggest the following:

1. To prevent confusion, the author chose KEGG instead of WikiPathways database for in vitro study using primary human macrophages. But for analysis of Gene Expression in PBMCs from RA and SLE patients, the author keeps using WikiPathways database. Justification for the selection of different database needs to be included in the discussion.

KEGG database does not include (Type I or II) Interferon pathways. Therefore, it is not suitable to use KEGG to analyze the RA/SLE patient data with consideration that the interferon signature in SLE patients will not be reflected. On the other hand, Wikipathways include interferon pathways, which fits RA/SLE analysis and thus justifies Wikipathways as an appropriate and solid method for RA/SLE pathway analysis. Nonetheless, the reviewer's suggestion is taken, and we have employed IMPaLA (Integrated Molecular Pathway Level Analysis), which combines more than ten databases including KEGG and Wikipathways, to perform pathway analysis again throughout the manuscript including Fig. 3a,b, Fig.4b, Fig.7a, Fig.8b,c (RA/SLE dataset). The results are very similar to those analyzed by either KEGG or Wikipathways, and do not affect our conclusion. We have replaced the data accordingly, and changed the pathway analysis method to IMPaLA (Integrated Molecular Pathway Level Analysis) on pp. 18 and 20.

2. The authors used LysMCre+ littermates as control. However, it was noted as WT in the figures and many other parts of the manuscript, which is incorrect. The control used need to be cleared properly noted.

We have stated in both Results (pg. 5) that 'Sex-matched *LysMcre*⁺ littermates served as wild type (WT) controls' and Methods (pg. 15) section that '...their littermates with *Tgfb β 2*^{+/+};*LysMcre*(+) genotype as WT controls were used for experiments'. According to the reviewer's comment, we further added 'referred to as WT' in these sentences on pp. 5 and 15.

We feel that the revised manuscript addresses the issues raised during review and the manuscript has been significantly strengthened. Thank you for considering the revised manuscript for publication in *Nature Communications*.